# PI(4,5)P₂ diffuses freely in the plasma membrane even within high-density effector protein complexes

Jonathan Pacheco[1] , Anna C. Cassidy[1] , James P. Zewe[1] , Rachel C. Wills[1] , and Gerald R.V. Hammond[1]

The lipid phosphatidyl-D-*myo*-inositol-4,5-*bis*phosphate [PI(4,5)P₂] is a master regulator of plasma membrane (PM) function. Its effector proteins regulate transport, signaling, and cytoskeletal processes that define PM structure and function. How a single type of lipid regulates so many parallel processes is unclear. We tested the hypothesis that spatially separate PI(4,5)P₂ pools associate with different PM complexes. The mobility of PI(4,5)P₂ was measured using biosensors by single-particle tracking. We found that PM lipids including PI(4,5)P₂ diffuse rapidly (∼0.3 μm²/s) with Brownian motion, although they spend one third of their time diffusing more slowly. Surprisingly, areas of the PM occupied by PI(4,5)P₂-dependent complexes did not slow PI(4,5)P₂ lateral mobility. Only the spectrin and septin cytoskeletons showed reduced PI(4,5)P₂ diffusion. We conclude that even structures with high densities of PI(4,5)P₂ effector proteins, such as clathrin-coated pits and focal adhesions, do not corral unbound PI(4,5)P₂, questioning a role for spatially segregated PI(4,5)P₂ pools in organizing and regulating PM functions.

## Introduction

The inner leaflet of an animal cell's plasma membrane (PM) is a bustling hub of transport, signaling, and structure. It is primarily here that cells regulate incoming and outgoing vesicular traffic, control selective permeability through channels and transporters, and facilitate ion and lipid exchange with the ER by maintaining membrane contact sites. The lipid bilayer maintains structural rigidity by attaching the underlying cortical cytoskeleton and builds adhesion complexes that enable cells to integrate into tissues. It also assembles numerous signal transduction complexes to relay extrinsic signals and modify cell function to meet organismal needs. Regulation of these diverse processes relies on proteins that are recruited to and/or activated at the PM by a single class of regulatory molecule: the lipid, PI(4,5)P₂ (Saarikangas et al., 2010; Schink et al., 2016; Hammond and Hong, 2018; Dickson and Hille, 2019; Hammond and Burke, 2020). Therefore, understanding the spatial distribution of PI(4,5)P₂ (phosphatidyl-D-*myo*-inositol-4,5-*bis*phosphate) in the PM, and how it couples to these manifold proteins, is essential to understanding PM function at large.

An attractive hypothesis has been that PI(4,5)P₂ is spatially segregated into pools that couple to specific PM functions; these functions can then be independently regulated through local changes in PI(4,5)P₂ concentration, either through lipid corralling or local metabolism (Gamper and Shapiro, 2007; Hammond, 2016). Indeed, specific enrichment of the lipid has been observed at sites of regulated exocytosis, caveolae, and clusters of actively

signaling K-Ras4B (van den Bogaart et al., 2011; Trexler et al., 2016; Zhou et al., 2015; Fujita et al., 2009; Zhou et al., 2021). However, in these cases, it is likely that PI(4,5)P₂-binding proteins are responsible for enriching the lipid; there is no evidence that a pre-existing local PI(4,5)P₂ pool recruits the proteins or that physiological changes in local PI(4,5)P₂ concentration modulate their function. Synthesis of PI(4,5)P₂ has been reported to localize specifically at lipid rafts (Johnson et al., 2008; Myeong et al., 2021). Could this underpin spatial control of individual PM functions? It is worth noting that transport, signaling, and cytoskeletal processes regulated by PI(4,5)P₂ occur in complexes that are hundreds of nanometers to microns in size, and happen over second- to minute-time scales. Lipid rafts, on the other hand, are nanoscopic and ephemeral structures in living cells, resolving over nanometer and millisecond scales (Levental et al., 2020).

If cells have the ability to form and maintain spatially segregated pools of PI(4,5)P₂, this must occur in the context of opposing diffusion of this molecule in the fluid environment of the PM. PI(4,5)P₂ diffusion has been found to be rapid, with a diffusion coefficient of 0.1–1 μm²/s in living cells (Mashanov and Molloy, 2007; Yaradanakul and Hilgemann, 2007; Golebiewska et al., 2008; Hammond et al., 2009). However, these measurements have been obtained from studies of bulk diffusion in the PM and may not detect reduced mobility in the macromolecular complexes driving PM function. Indeed, diffusion of

[1]Department of Cell Biology, University of Pittsburgh School of Medicine, Pittsburgh, PA.

Correspondence to Gerald R.V. Hammond: ghammond@pitt.edu.

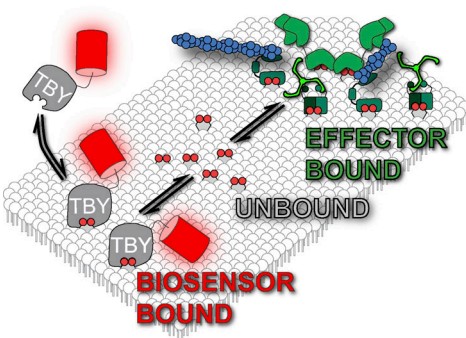

**Figure 1. Lipid biosensors and pools of PM lipid.** Functional membrane lipids such as PI(4,5)P$_2$ are expected to exist in a dynamic equilibrium between "unbound lipid" where the headgroup does not engage proteins, and "effector bound" pool where the headgroup binds effector proteins. Biosensors like Tubby$_c$-PAmCherry (TBY) reversibly interact with and thus sample the unbound pool of lipid.

extracellular lipids has been observed to be slowed by the cortical cytoskeleton's attachment to the PM (Fujiwara et al., 2002; Andrade et al., 2015; Fujiwara et al., 2016). We, therefore, tested the hypothesis that PI(4,5)P$_2$ diffusion is reduced at macromolecular complexes that drive specific PM functions. To this end, we employed single-particle tracking photoactivation localization microscopy (sptPALM; Manley et al., 2008) to measure the diffusion of PI(4,5)P$_2$ in living cells. We determine diffusion at PI(4,5)P$_2$-dependent macromolecular complexes, labeled by expression of fluorescently tagged transgenes or incorporation of such tags onto endogenous proteins by gene editing. We report that, for the most part, free PI(4,5)P$_2$ diffuses unhindered inside and between such complexes. However, PI(4,5)P$_2$ diffusion is substantially reduced in regions of the membrane that are highly enriched with spectrin or septin filaments, which are components of the cortical cytoskeleton that integrate tightly with the membrane.

## Results

### Diffusion of inner leaflet PI(4,5)P$_2$ and other lipids measured by sptPALM using genetically encoded lipid biosensors

PI(4,5)P$_2$ in the PM exists in rapid dynamic equilibrium with its effector proteins (Fig. 1). Estimates using fluorescent acyl chain derivatives indicate that two out of three PI(4,5)P$_2$ molecules are in complex with such proteins at any given moment (Golebiewska et al., 2008). In this manuscript, we consider the remaining one third of lipid molecules that are estimated to be unbound. When new complexes are assembled, or new effector proteins are recruited to these complexes, it is this unbound lipid that recruits them; this is the lipid pool that must be locally concentrated to modulate an effector complex. We, therefore, measured the diffusion of these unbound lipid molecules. To this end, we used genetically encoded lipid biosensors that interact with the headgroup. The biosensors themselves are in rapid dynamic equilibrium with the lipids and preclude interaction with endogenous effector proteins (Fig. 1). Unlike effector proteins, the biosensors are estimated to sequester a much smaller

fraction of PI(4,5)P$_2$, likely <10% (Wills et al., 2018). Although expression of high concentrations of biosensors can sequester a higher fraction of the unbound pool, reducing the pool available for effector interaction and inhibiting phospholipase C, for example Várnai and Balla (1998), this is unlikely to occur in the experiments described below since low expression levels were utilized to favor resolution of single molecules. Crucially, previous studies have shown that the diffusion coefficient of biosensor-bound PI(4,5)P$_2$ is unchanged from the unbound lipid (Mashanov and Molloy, 2007; Yaradanakul and Hilgemann, 2007; Golebiewska et al., 2008; Hammond et al., 2009).

To obtain local diffusion coefficients in intact PM of living HeLa cells, we employed sptPALM (Manley et al., 2008). In this approach, a photoactivatable fluorescent protein is switched on with low intensities of the activating wavelength (in our case, PAmCherry1 with 405 nm light), sufficient to generate sparse and resolvable single fluorescent molecules on the ventral PM, when viewed by total internal reflection fluorescence microscopy (TIRFM; Fig. 2 A). Subsequent activation with high intensities of 405 nm light leads to activation of a large population of molecules, which are no longer resolvable. This leads to the characteristic, uniform sheet of PM fluorescence as seen with nonactivatable fluorescent protein conjugates in TIRFM (Fig. 2 B). To confirm that single fluorescent puncta were indeed single molecules, we applied the stringent DISH criteria: "(1) diffraction-limited size, (2) intensity of emission appropriate for a single fluorophore, (3) single-step photobleaching, and (4) half-life of the fluorophore population before photobleaching occurred, [inversely] proportional to laser excitation power" (Mashanov et al., 2004). Indeed, our fluorescent spots had (1) a diffraction-limited size with a mean of 201 nm, consistent with the expected 206 nm size at 596 nm peak emission when imaged through our 1.45-NA objective lens (Fig. 2 C); (2) an intensity distribution that is log-normal with a mean of 470 photons (Fig. 2 D), consistent with prior measurements of PAmCherry1 (Subach et al., 2009); (3) single-step photobleaching (Fig. 2 E); and (4) a half-life time before photobleaching that was inversely proportional to excitation power (Fig. 2 F). Thus, we were able to detect single biosensor:lipid complexes in the PM. We did note a minor peak of fluorescence spot size centered around the 65–75 nm bin (Fig. 2 C), representing <2.5% of all localizations. Given our pixel size of 65 nm in image space, we believe this is caused by "hot pixels" in our sCMOS that locally enhance signal and lead to an artifactually narrower distribution in a few single molecule localizations.

The single biosensor molecules represented complexes with lipids on the membranes imaged in TIRFM. They are readily discerned from unbound, cytoplasmic biosensors diffusing just above the membrane but in the plane of illumination by the relative diffusion rates: lipids diffuse up to ~1 μm²/s in cells, whereas cytosolic biosensors diffuse around 20 μm²/s (Hammond et al., 2009). Displacement of such molecules in the camera exposure time, t = 55 ms, can be estimated from $\sqrt{4Dt/\pi}$ (Teruel and Meyer, 2000) at ~260 nm and 1.2 μm, respectively. The large degree of displacement in the latter case "blurs" and spreads the intensity of the single molecule image to the extent that it is no longer resolvable against camera noise. It follows

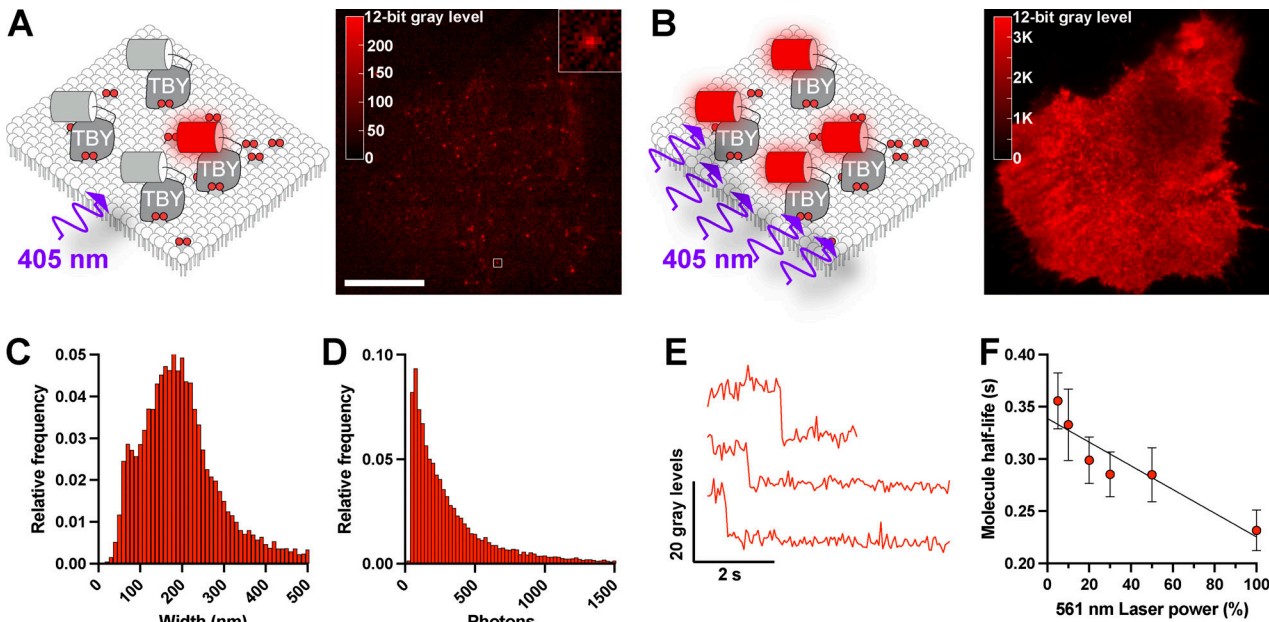

Figure 2. **Single-molecule detection of lipid biosensors in the PM. (A)** Single particle tracking PALM using PAmCherry-tagged lipid biosensors. Illumination of a HeLa cell expressing PAmCherry-Tubby$_c$ (TBY) PI(4,5)P$_2$ biosensor with low-intensity 405 nm light activates few fluorescent proteins that can be resolved as individual fluorescent spots. Scale bar = 20 µm, inset is 2 µm. **(B)** Activation of PAmCherry with high-intensity 405 nm illumination activates the majority of PAmCherry-Tubby$_c$, revealing the overall PM distribution in the same HeLa cell as B. **(C and D)** Fitted diameter (C) and photon count (D) of 14,095 individual spots from a representative cell detected with Thunderstorm. **(E)** Example fluorescence intensity profiles of individual spots showing single-step photobleaching. **(F)** Half-life of spot trajectories in time-lapse images measured at varying power modulation of the 561 nm excitation laser, showing half-life decreases proportionally with excitation power; data are grand means ± SE from time-lapse recordings of seven cells.

that biosensors can only be tracked for the lifetime that they stay in complex with the lipid. Estimates of the lifetime of Tubby$_c$ molecules extrapolated back to zero laser-induced photobleaching place this binding lifetime at 339 ms (Fig. 2 E). Enzymatic turnover of the lipids or engagement with an endogenous effector protein requires the dissociation of the biosensor first, so these processes do not impact our diffusion measurements. Indeed, we have previously demonstrated that rates of biosensor-bound lipid catabolism are limited by the dissociate rate of the biosensor (Hammond et al., 2009).

We performed time-lapse imaging of these complexes at ~18 frames per second and employed posthoc tracking analysis using the versatile and accurate single-molecule tracking algorithm, TrackMate (Tinevez et al., 2017). This algorithm defines single-molecule trajectories from which we analyzed two properties: the turning angle, θ, between two steps in a trajectory (disregarding direction, giving a range of 0–180°) and the displacement of localizations in the trajectory over increasing time lags (Fig. 3 A). We used a variety of lipid biosensors: two for PI(4,5)P$_2$, the pleckstrin homology (PH) domain from phospholipase C (PLC) δ1, PH-PLC (Várnai and Balla, 1998), and the c-terminal domain from Tubby, Tubby$_c$ (Quinn et al., 2008); two for the PI(4,5)P$_2$ precursor PI4P, which were the PI4P binding domains of *Legionella* effector proteins SidM, P4Mx2 (Hammond et al., 2014) and SidC, P4C (Weber et al., 2014); and one for the abundant inner leaflet phospholipid, phosphatidylserine, namely the C2 domain from lactadherin, Lact-C2 (Yeung et al., 2008). We also employed the myristoylated (C$_{14}$) and palmitoylated (C$_{16}$) 11-residue peptide from Lyn kinase, Lyn$_{11}$ (Teruel et al.,

1999). This enabled us to generate a more generalizable measurement of lipid diffusion on the inner leaflet of the PM.

For a particle exhibiting true Brownian motion, i.e., unconstrained diffusion, displacement is random, leading to an even distribution of turning angle θ, with a mean of 90° (Burov et al., 2013). As exemplified for the Tubby$_c$ PI(4,5)P$_2$ biosensor, we observed θ distributions that were close to random, with a slight tendency toward more obtuse angles (Fig. 3 B). All our probes exhibited a small but significant increase in turning angle from the predicted 90° (between 97° and 102°, Fig. 3 C and Table 1). Therefore, diffusion seemed almost random, but there was a small tendency to reflect back in the direction of motion, perhaps indicating collision with immobile obstacles (Burov et al., 2013). Considering the displacement of the molecules, their mean square displacement increased largely linearly with increasing time lag (Fig. 3 D); the slope of this line defines the diffusion coefficient (Einstein, 1905). The non-PI(4,5)P$_2$ biosensors exhibited remarkably similar diffusion coefficients of ~0.3 µm²/s (Fig. 3 E). For PI(4,5)P$_2$, PH-PLCδ1 was about half that at 0.14 µm²/s (95% confidence interval [C.I.] 0.12–0.17) and Tubby$_c$ somewhat intermediate at 0.23 µm²/s (95% C.I. 0.17–0.29). ANOVA revealed a consistently significant difference of PH-PLCδ1 to the other probes, whereas Tubby$_c$ was not consistently significantly different from the other probes (Table 2). Therefore, we can conclude that the PH-PLCδ1 biosensor diffuses slower than the other lipid probes, but this is not clear for Tubby$_c$. Notably, our previous work showed that impeded diffusion of the PH-PLCδ1 protein is not a function of its lipid binding properties and likely represents an additional,

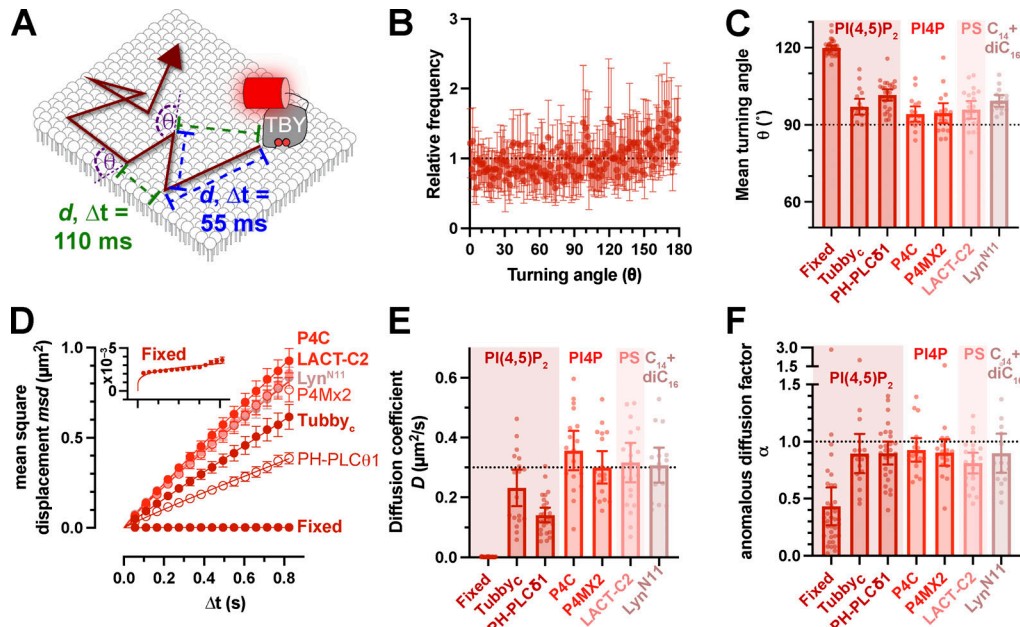

Figure 3. **Rapid Brownian diffusion of lipid biosensors in the PM. (A)** Single particle tracking PALM using PAmCherry-tagged lipid biosensors produces trajectories with measured displacements $d$ between localizations separated by various camera exposure-derived time lags, $\Delta t$. The turning angle θ between successive displacements can also be measured. **(B and C)** The distribution of turning angles measured from 18 cells expressing Tubby$_c$-PAmCherry (means ± 95% C.I.), which is more conveniently represented in C as the mean turning angle, is close to 90° (no bias in turning angle) for a variety of lipid biosensors. **(D)** These lipid biosensors display a linear increase in mean square displacement over time, indicating Brownian motion. Fit is to $msd = 4D\Delta t^{\alpha}$. **(E and F)** Mean diffusion coefficients (E) and anomalous diffusion factor, α (F), from individual HeLa cells for the indicated lipid biosensors are shown. The inset for D shows a zoomed axis for the fixed cell data. Data in C–F are grand means ± 95% C.I. of trajectories from 18 (Tubby$_c$, P4C, and P4Mx2), 27 (PH-PLCδ1), 22 (Lact-C2), 16 (Lyn$^{N11}$), or 35 (Tubby$_c$-fixed) cells.

undefined interaction(s) that impedes its diffusion (Hammond et al., 2009). On the other hand, our estimates of inner leaflet lipid diffusion here are consistent with prior estimates for PI(4,5)P$_2$ that ranged from 0.1 to 1 μm²/s (Mashanov and Molloy, 2007; Yaradanakul and Hilgemann, 2007; Golebiewska et al., 2008; Hammond et al., 2009).

For freely diffusing particles, the apparent diffusion coefficient $D$ is constant at all observed time lags. However, impediments to free diffusion can cause the apparent $D$ to decrease at increasing time lags, leading to a downward curve in the mean square displacement vs. time plot; this curve is described by the exponent α, where α = 1 describes free diffusion and α < 1 describes impeded or anomalous diffusion (Saxton, 1994). Our range of measurements for α was close to 1 for all probes (Fig. 3 F), for the most part averaging ≥0.9, which is considered Brownian motion. The one exception was Lact-C2 at α = 0.81 (95% C.I. = 0.72–0.90), which is significantly reduced from 1 (one sample $t$ test, hypothesized mean = 1, t = 3.524, P = 0.002). Nonetheless, deviations from Brownian motion tended to be the

exception for these lipid biosensors, indicating largely free Brownian motion in the inner leaflet of the PM.

As a control for the precision of our single particle tracking experiments, we fixed cells expressing Tubby$_c$ with glutaraldehyde to immobilize the biosensor protein. Analysis of time-lapse data from such cells revealed a highly skewed distribution of turning angle θ toward obtuse angles (Fig. 3 C) and a very slow diffusion coefficient (Fig. 3, D and E) of 0.0013 μm²/s that was highly anomalous (α = 0.43, 95% C.I. = 0.27–0.60; one sample $t$ test compared with the hypothesized value of 1; t = 61.85, P < 0.0001). This result was not derived from slow diffusion of the fixed probe; rather, it was driven by a mean square displacement at an of average 0.0027 μm² across all time lags (see inset of Fig. 3 D). This corresponds to a constant displacement of √0.0027 μm² = 52 nm, representing the precision of our single-molecule localization measurements. This places a limit of lateral resolution on our diffusion measurements of ~100 nm by the Nyquist sampling theorem.

Table 1. **One sample, two-tailed $t$ test of turning angle θ compared to a theoretical mean of 90°**

| Biosensor: | Fixed | Tubby$_c$ | PH-PLCδ1 | P4C | P4MX2 | LACT-C2 | Lyn$^{N11}$ |
|---|---|---|---|---|---|---|---|
| $n$ | 35 | 18 | 27 | 18 | 18 | 22 | 16 |
| $t$ | 61.85 | 4.814 | 10.88 | 2.925 | 2.416 | 3.524 | 8.506 |
| P (two-tailed) | **<0.0001** | **0.0002** | **<0.0001** | **0.0095** | **0.0273** | **0.0020** | **<0.0001** |

Data from Fig. 3 C. Significant results are highlighted in bold. n is the number of cells.

Table 2. **P values from Tukey's multiple comparison test for biosensor diffusion coefficients presented in Fig. 3 E**

| Biosensor: | Fixed | Tubby$_c$ | PH-PLCδ1 | P4C | P4MX2 | LACT-C2 |
|---|---|---|---|---|---|---|
| Tubby$_c$ | **<0.0001** | | | | | |
| PH-PLCδ1 | **<0.0001** | 0.0558 | | | | |
| P4C | **<0.0001** | **0.0054** | **<0.0001** | | | |
| P4MX2 | **<0.0001** | 0.3916 | **<0.0001** | 0.6421 | | |
| LACT-C2 | **<0.0001** | 0.1221 | **<0.0001** | 0.8735 | 0.9989 | |
| Lyn$^{N11}$ | **<0.0001** | 0.3050 | **<0.0001** | 0.8001 | >0.9999 | >0.9999 |

Significant variation was observed among groups by one-way ANOVA (F = 43.14, P < 0.0001). Significant results are highlighted in bold.

Analyzing diffusion based on whole trajectories gives a mean diffusion coefficient for that particle, but it does not take into account potential changes in $D$ as the molecule encounters obstacles that slow its motion. Indeed, visual inspection of our trajectories seems to show different classes: rapid diffusers with large displacements, slow diffusers with short displacements, and a mixture of large and small displacements, which represented the majority (Fig. 4 A). We, therefore, took an alternative approach to define $D$ at the population level. We pooled displacements across entire populations from single cells at specific time lags (e.g., 220 ms, as shown in Fig. 4 B). Plotting the cumulative distribution allows a fit of the apparent diffusion coefficient (Vrljic et al., 2002). Fitting a single population of diffusers revealed the experimental population had a higher number of shorter displacements and fewer longer displacements than predicted (Fig. 4 B, dashed line). Assuming two populations, a fast and a slow, yielded a much tighter fit to the data (Fig. 4 B, solid line). All the lipid biosensors exhibited similar distributions with ~30% of displacements representing slow diffusion and ~70% being fast, except PH-PLCδ1, which had a 50/50 split (Fig. 4 C). The fast population $D$ ranges from 0.3 to 0.7 µm²/s, which is still consistent with the range estimated for PI(4,5)P$_2$ diffusion (Mashanov and Molloy, 2007; Yaradanakul and Hilgemann, 2007; Golebiewska et al., 2008; Hammond et al., 2009), whereas the slow population $D$ is <0.05 µm²/s (Fig. 4 D). Performing this analysis across the first four time lags allows the time-dependence of $D$ to be interrogated (Fig. 4 E). Across all biosensors, the fast population exhibited diffusion coefficients that were time lag independent, i.e., $\alpha = 1$ or very close to this value (Fig. 4 F). On the other hand, the slow populations of PH-PLCδ1, Lact-C2, and Lyn$_{11}$ exhibited $\alpha \leq 0.7$, which is significantly different from 1, indicating some degree of anomaly, i.e., hindered diffusion (Table 3).

Therefore, it appears that most free lipids, including PI(4,5)P$_2$, exhibit rapid, unhindered Brownian motion on the inner leaflet of the PM. A minor population exhibits slower and potentially hindered diffusion. We next turned our attention to whether this free and hindered diffusion was associated with specific PI(4,5)P$_2$-dependent macromolecular complexes in the PM.

## Diffusion of PI(4,5)P$_2$ biosensor at PI(4,5)P$_2$-dependent effector complexes

For these experiments, we elected not to employ the PH-PLCδ1 since we found in the previous section that it diffused

significantly slower than other lipid biosensors, which can be ascribed to lipid-extrinsic protein–protein interactions (Hammond et al., 2009). We instead employed the Tubby$_c$ PI(4,5)P$_2$ biosensor since Tubby$_c$ behaves most similarly with the other lipid sensors (Figs. 3 and 4; and Table 3). This is a good indication of unimpeded diffusion of the lipid:biosensor complex, since a lipid-selective biosensor interacts with the headgroup of the lipid, preventing lipid-selective interactions with other proteins that could impair diffusion for the duration of the biosensor complex. Diffusion should therefore be mainly limited by the viscous drag of the acyl chains in the membrane, which is not expected to differ significantly among different lipid classes (or the dually acylated Lyn$_{11}$ peptide). For Tubby$_c$, but not PH-PLCδ1, this seems to be the case.

We performed sptPALM with PAmCherry1-Tubby$_c$ in HeLa cells expressing EGFP- or sfGFP-conjugated markers of specific PI(4,5)P$_2$-dependent PM macromolecular complexes. We then segment trajectories based on whether they contact the domains, defined by thresholding of the fluorescence intensity (see Materials and methods for details). We then compare diffusion inside these domains to outside at the single-cell level. As we described in the previous section, expression of Tubby$_c$-mCherry is unlikely to sequester a substantial fraction of the unbound PI(4,5)P$_2$ and compete with PI(4,5)P$_2$-dependent macromolecular complexes, especially when expressed at low levels to facilitate single-molecule detection. To verify that this was true, we compared such complexes in the presence and absence of Tubby$_c$-mCherry, revealing no apparent changes as shown in Fig. S1.

Although single-molecule localization is super-resolution (estimated at ~100 nm in our experiments), the GFP signal is still subject to the Raleigh limit of lateral resolution when viewed by TIRFM. The threshold-defined domains are therefore convolved with the point-spread function of GFP, appearing slightly larger than they in fact are. For this reason, we interpret our data specifically to interrogate diffusion inside and in close vicinity of these structures. Nonetheless, as will be seen by the numerous examples depicted in the following, trajectories explored the full area of the threshold-defined domains and were not restricted to their periphery. So, although the point spread function "blurring" of the domains' periphery may skew the data to include some "outside" diffusion behavior as "inside," changes within the domains will still be detected and all but the subtlest of changes will be measured.

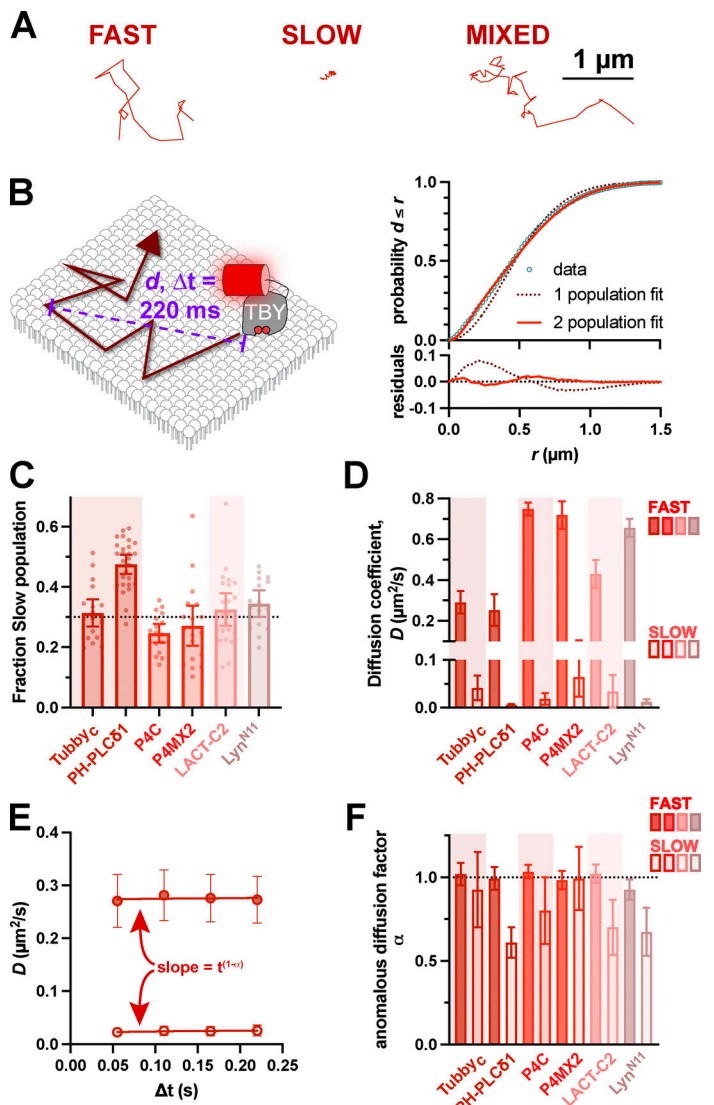

**Figure 4.** **Existence of both fast and slow diffusing lipid molecules in the PM. (A)** Examples of representative single-molecule trajectories, showing either fast-moving, slowly moving, or mixed trajectories. **(B)** Separating trajectories from all molecules into distinct radial displacements (*d*) at defined Δ*t* (e.g., 220 ms) allows *D* to be estimated from the distribution of *d* values independently of individual (often mixed) trajectories. This distribution is much more tightly estimated by assuming two populations, one fast and one slow. **(C)** Fraction of radial displacements assigned to the slowly diffusing population. **(D)** Mean diffusion coefficients for both fast and slow populations of each lipid biosensor. **(E)** By comparing *D* values from the distribution of radial displacements at different Δ*t* values (i.e., 55, 110, 165, and 220 ms), the dependence of *D* on time interval can be estimated as the slope $t^{1-\alpha}$, where α = 1 reveals no change and α < 1 indicates decreasing apparent *D* with time. Data are from the Tubby$_c$ biosensor and are grand means ± 95% C.I. (18 cells). **(F)** Anomalous diffusion factor α for both fast (closed) and slow (open symbols) for each biosensor. Only slowly diffusing molecules show evidence of anomalous diffusion. For (B, C, and E), data are grand means ± 95% C.I. of measurements from the same 18 (Tubby$_c$, P4C, and P4Mx2), 27 (PH-PLCδ1), 22 (Lact-C2), or 16 (Lyn$^{N11}$) cells as shown in Fig. 2.

The first structure that we considered was ER–PM contact sites (Fig. 5 A). Here, ER membranes are anchored in proximity to the PM to facilitate ion and lipid exchange between the organelles (Wu et al., 2018). Most tethering factors identified to date utilize an interaction with PI(4,5)P$_2$ for PM attachment (Giordano et al., 2013; Sohn et al., 2018; Besprozvannaya et al., 2018). We selected one of the most abundant tethering factors, extended synaptotagmin 1 (E-Syt1), tagged with sfGFP on an endogenous allele, allowing us to observe endogenous ER–PM contact sites (Zewe et al., 2018). Endogenous E-Syt1 exhibits a punctate morphology, often strung along tubule-like distributions (Fig. 5 B); the size and density of such structures were not altered by Tubby$_c$ expression (Fig. S1 A). These structures were frequently crisscrossed by PI(4,5)P$_2$:Tubby$_c$ complexes. Surprisingly, we found no changes in diffusion coefficients, α, turning angle, or the fraction of the population exhibiting slow diffusion (Fig. 5 C). In short, E-Syt1–defined ER:PM contact sites seemed to offer no impediment to free PI(4,5)P$_2$ diffusion.

Table 3. **One sample, two-tailed *t* test of anomalous diffusion factor α, compared to a theoretical mean of 1**

| Biosensor | Tubby$_c$ | | PHPLCδ1 | | P4C | | P4MX2 | | LACT-C2 | | Lyn$^{N11}$ | |
|---|---|---|---|---|---|---|---|---|---|---|---|---|
| | Fast | Slow | Fast | Slow | Fast | Slow | Fast | Slow | Fast | Slow | Fast | Slow |
| *n* | 18 | | 27 | | 17 | | 17 | | 22 | | 16 | |
| *t* | 0.5999 | 0.6809 | 0.1799 | 8.787 | 1.798 | 2.119 | 0.5880 | 0.06272 | 0.8254 | 3.756 | 2.546 | 4.831 |
| *P* | *0.5565* | *0.5051* | *0.8586* | **<0.0001** | *0.0911* | *0.0501* | *0.5647* | *0.9508* | *0.4184* | **0.0012** | **0.0224** | **0.0002** |

Data from Fig. 4 F. Significant results are highlighted in bold.

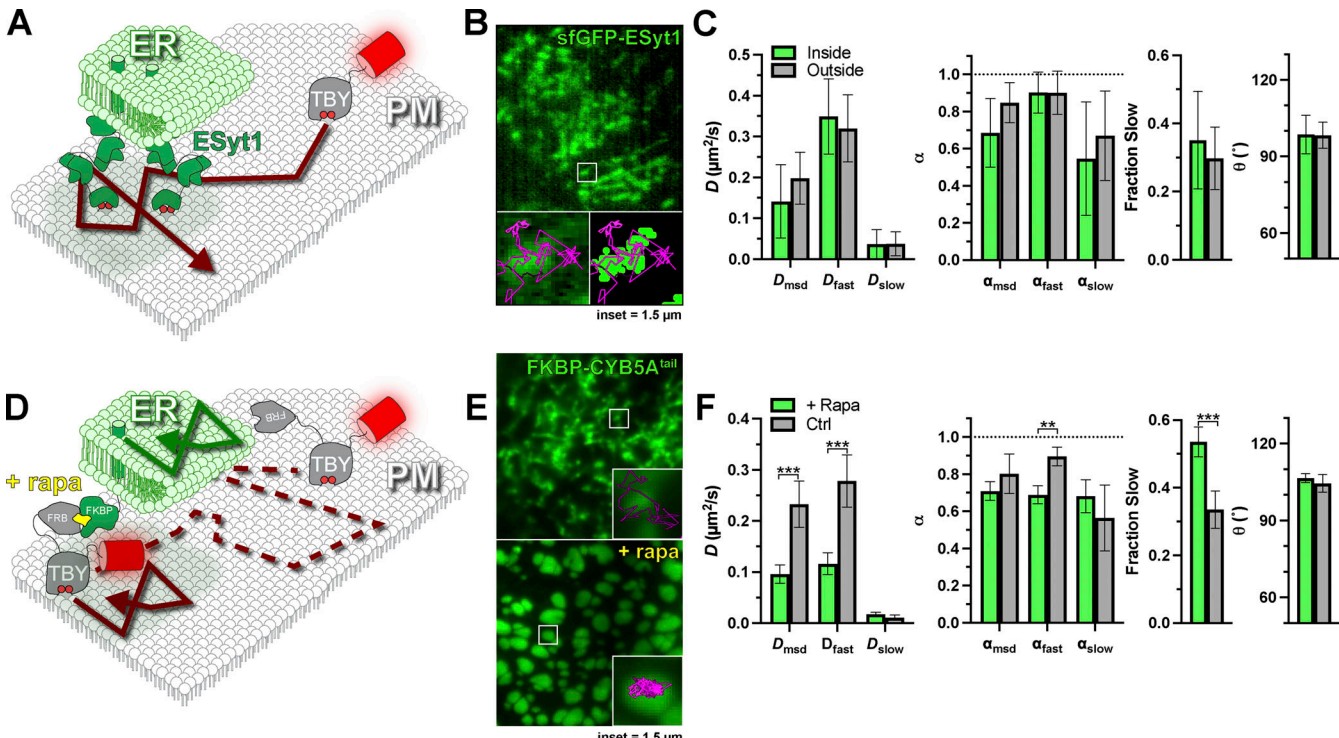

Figure 5. **PI(4,5)P₂ diffusion is uninhibited at ER–PM contact sites. (A)** Schematic of diffusion in the PM proximal to ER–PM contact sites marked with endogenously tagged sfGFP-ESyt1. **(B)** TIRF image of a small region of PM from an edited sfEGFP-ESyt1 cell; insets show a region of raw and thresholded images overlaid with single-molecule trajectories. Inset = 1.5 µm. **(C)** Diffusion coefficients, α (from msd v Δt plots as well as fast and slow populations), the fraction of the population with slow diffusion, and mean turning angles are shown for trajectories classified as inside or outside ER–PM contact sites. Data are the grand means ± 95% C.I. of 11 cells. **(D)** FRB-Tubby_c PI(4,5)P₂ biosensor can be forced into ER–PM contact sites by rapamycin (rapa)-induced dimerization with ER-resident FKBP-CYB5A^tail. **(E)** Images show EGFP-FKBP-CYB5A^tail before and after induction of ER–PM contact sites by rapa-induced dimerization with FRB-Tubby_c-PAmCherry. Insets show a region overlaid with single molecule trajectories. Inset = 1.5 µm. **(F)** Diffusion coefficients, α (from msd v Δt plots as well as fast and slow populations), the fraction of the population with slow diffusion, and mean turning angles are shown for separate cells treated with or without rapa. Data are grand means ± 95% C.I. of 28 (+rapa) or 19 (control) cells. **P ≤ 0.01, ***P ≤ 0.001 from paired *t* test with Holm–Šidák correction for multiple comparisons.

To establish that we were able to detect changes in diffusion at ER–PM contact sites, we aimed to trap Tubby_c at ER–PM contact sites using chemically induced dimerization. To this end, we tagged PAmCherry1-Tubby_c with the FKBP-rapamycin binding (FRB) domain of mTor. We could then recruit this fusion to contact sites using rapamycin to induce FRB dimerization with FK506 binding protein (FKBP) fused to the ER-localized tail of CYB5A (Zewe et al., 2018), as shown in Fig. 5 D. When viewed in TIRFM, eGFP-tagged FKBP–CYB5A^tail showed characteristic tubular ER morphology before rapamycin addition (pseudocolored green in Fig. 5 E) but formed large puncta, characteristic of induced ER–PM contact sites upon rapamycin addition (Fig. 5 E). Whereas trajectories of Tubby_c crisscrossed CYB5A-labeled tubules before rapamycin addition, they became trapped within the puncta upon rapamycin addition (Fig. 5 E), as expected (Fig. 5 D). Notably, we observed substantial changes in diffusion under these conditions. Upon trapping at contact sites, diffusion halved when assessed either by trajectory mean square displacements or the fast component of the entire population, and the fast population became slightly, but significantly, more anomalous (Fig. 5 F). The fraction of displacements exhibiting slow diffusion also increased from ∼30–50% (Fig. 5 F). On the other hand, diffusion of the slow population was not affected,

nor was the distribution of turning angles (Fig. 5 F). In a sense, this result was surprising, since trapping inside a domain curtails particle displacements above the domain size and causes the particles to rebound as they encounter the boundary, increasing the number of obtuse turning angles (Burov et al., 2013). The reason that we did not observe this behavior is likely the large, micron-sized contact sites that were induced (Fig. 5 E). Even for the fast population of Tubby_c diffusing at ∼0.1 µm²/s imaged across our typical analysis window of 0.22 s, average total displacement will only be 0.17 µm—estimated from $\sqrt{4Dt/\pi}$, where t is the time lag (Teruel and Meyer, 2000). Therefore, the duration of our fluorescence tracking (limited by photobleaching, Fig. 2 F) is just below that needed for the edge effects of the large domains to become apparent, explaining why α is only significantly reduced for the fast population (Fig. 5 F).

We next turned our attention to clathrin-containing structures (CCS). Clathrin-mediated endocytosis is reliant on the PI(4,5)P₂-dependent recruitment of cargo adapter proteins and fission machinery to build and bud an endocytic vesicle (Mettlen et al., 2018). As a result of this dense, tightly membrane-associated complex, diffusion of unbound lipids has been proposed to be greatly reduced (Schöneberg et al., 2017). We, therefore, interrogated Tubby_c diffusion in and around CCS by

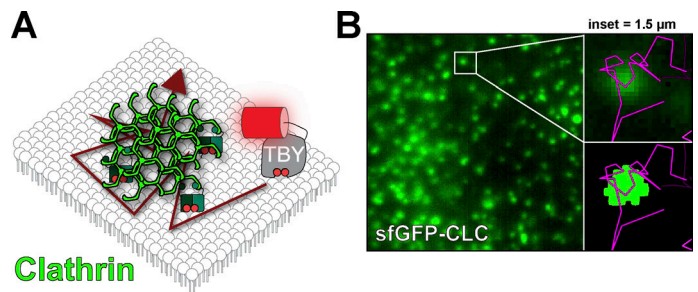

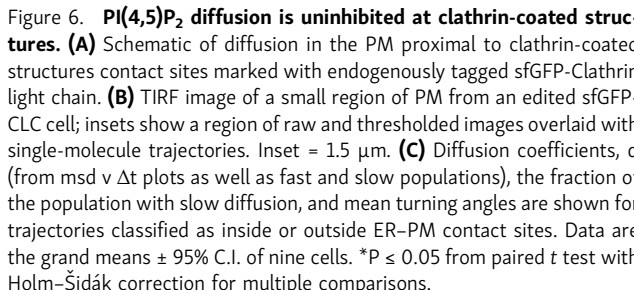

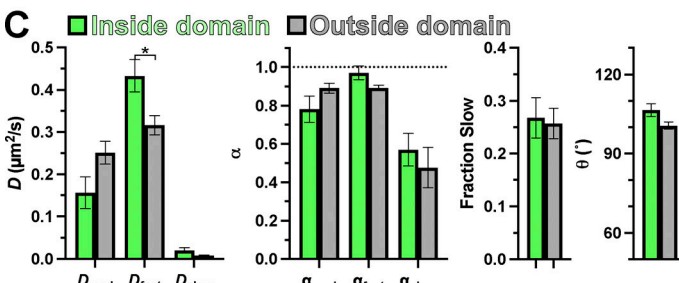

**Figure 6. PI(4,5)P₂ diffusion is uninhibited at clathrin-coated structures. (A)** Schematic of diffusion in the PM proximal to clathrin-coated structures contact sites marked with endogenously tagged sfGFP-Clathrin light chain. **(B)** TIRF image of a small region of PM from an edited sfGFP-CLC cell; insets show a region of raw and thresholded images overlaid with single-molecule trajectories. Inset = 1.5 μm. **(C)** Diffusion coefficients, α (from msd v Δt plots as well as fast and slow populations), the fraction of the population with slow diffusion, and mean turning angles are shown for trajectories classified as inside or outside ER–PM contact sites. Data are the grand means ± 95% C.I. of nine cells. *P ≤ 0.05 from paired $t$ test with Holm–Šidák correction for multiple comparisons.

tagging endogenous clathrin light chain with sfGFP (Cho et al., 2021 *Preprint*; Leonetti et al., 2016). In these cells, CCS appear as diffraction-limited spots; the density and dynamics of these spots are not altered by Tubby$_c$ expression (Fig. S1 B). Biosensor trajectories passed apparently uninterrupted through these CCS (Fig. 6, A and B). Indeed, we did not measure any reduction in diffusion coefficient or α, nor changes in the turning angle or the fraction of molecules exhibiting slow diffusion (Fig. 6 C and Table 4). Thus, at least for the unbound fraction of PI(4,5)P₂ at CCS, the assembled clathrin lattice and its network of adaptor proteins do not present a measurable barrier to free diffusion.

One PM structure that has been proposed to have profound impacts on lipid diffusion (at least in the outer PM leaflet) is the cortical actin cytoskeleton (Morone et al., 2006; Andrade et al., 2015; Fujiwara et al., 2016). We labeled the endogenous F-actin cytoskeleton with EGFP-Lifeact (Riedl et al., 2008), revealing a dense array of cortical filaments in TIRFM (Fig. 7 A). These are thought to be bundled filaments rather than the meshwork of individual filaments that make up the majority of the cortical or membrane cytoskeleton (Morone et al., 2006). Tubby$_c$ crisscrossed these filaments without noticeable impediment, and we saw no change in any measured parameter to indicate non-Brownian motion in proximity of these filaments (Fig. 7 A).

There is no reason to suspect a specific interaction of PI(4,5)P₂ with bundled actin filaments. On the other hand, the lipid is intimately involved in activating proteins that attach the F-actin cytoskeleton to the membrane (Saarikangas et al., 2010), so we decided to measure PI(4,5)P₂ diffusion in proximity to such complexes. We selected three candidates: firstly, we considered ezrin-radixin-moesin (ERM)-proteins, which anchor actin filaments to the membrane in a PI(4,5)P₂-dependent manner (Algrain et al., 1993; Senju et al., 2017); these were labeled by expressing Ezrin-EGFP (Fig. 7 B). Secondly, we studied focal adhesions, whose assembly is thought to be driven by local PI(4,5)P₂ synthesis (Legate et al., 2011), activating proteins such as vinculin (Chinthalapudi et al., 2014); this was labeled by

integrating a split sfGFP tag at an endogenous *VCL* allele, yielding the expected 143K sfGFP-tagged vinculin protein (Fig. 7 C). Thirdly, we considered the branched F-actin nucleating complex Arp2/3, which relies on PI(4,5)P₂ for PM recruitment (Zoncu et al., 2007); and we imaged Arp2/3 by expressing a tagged component, ARPC4-EGFP (Fig. 7 D), which exhibits a punctate PM distribution when imaged by TIRFM (Zoncu et al., 2007), reminiscent of endogenously tagged ARPC3 and -4 in HEK293 cells (Cho et al., 2021 *Preprint*). There was no detectable change in the morphology of any of these structures with Tubby$_c$ expression (Fig. S1, C–E). To our surprise, we detected no change in Tubby$_c$ diffusion in and around ERM proteins or focal adhesions (Fig. 7, B and C; and Table 4). For Arp2/3, we measured a significant decrease in $D$ and α measured by mean square displacement (Fig. 7 D and Table 4). This was not reflected by changes in $D$ or α for the fast or slow components of diffusion, although there was a roughly 10% increase in the fraction of Tubby$_c$ molecules exhibiting slow diffusion (Fig. 7 D and Table 4), explaining the mean square displacement result. Collectively, these data do not reveal a substantial impact of the F-actin cytoskeleton on PI(4,5)P₂ diffusion at the temporal and spatial scales investigated herein—despite a small impediment evident at sites of branched F-actin nucleation by Arp2/3.

We next turned our attention to the nonactin components of the cortical cytoskeleton. Spectrins assemble into membrane-proximal filaments that integrate with the F-actin cortex (Bennett and Lorenzo, 2016). Unlike F-actin, these filaments are directly anchored to the PM in a PI(4,5)P₂-dependent manner (Wang and Shaw, 1995), forming a potential barrier to diffusion (Fig. 8 A). We labeled the spectrin cytoskeleton by expressing EGFP-β-spectrin, yielding the expected cortical distribution in HeLa cells when viewed by a confocal microscope (Fig. 8 B). In TIRFM, a largely amorphous but patchy distribution is observed (Fig. 8 C), similar to previous observations in fibroblasts (Ghisleni et al., 2020). This distribution is not altered by Tubby$_c$ expression (Fig. S1 F). Thresholding of this signal, therefore,

Table 4. **Results of paired t tests of the indicated mobility parameters inside and outside domains (or before and after rapamycin addition for FRB-Tubby$_c$) with the Holm–Šídák correction for multiple comparisons.**

| Domain: | | ER–PM contact sites | ER–PM contact sites | Clathrin-coated structures | F-actin | Cortical F-actin | Focal adhesions | Branched F-actin | Spectrin | Septins |
|---|---|---|---|---|---|---|---|---|---|---|
| Marker: | | E-Syt1 | FRB-Tubby$_c$ | CLC | Lifeact | Ezrin | Vinculin | ARPC4 | β-spectrin | Septin-2 |
| Fig: | | 5A | 5B | 6 | 7 | 7 | 7 | 7 | 8 | 9 |
| $D_{msd}$ | t-ratio | 2.165 | 4.765 | 3.361 | 3.029 | 1.463 | 2.38 | 3.482 | 3.445 | 5.207 |
| | df | 10 | 18 | 9 | 10 | 13 | 10 | 14 | 15 | 13 |
| | P | *0.3301* | ***0.0009*** | *0.0571* | *0.0856* | *0.5546* | *0.2700* | ***0.0281*** | ***0.0215*** | ***0.0014*** |
| $D_{fast}$ | t-ratio | 1.512 | 5.161 | 3.721 | 3.656 | 1.775 | 1.231 | 0.5662 | 1.891 | 3.309 |
| | df | 10 | 18 | 9 | 10 | 13 | 10 | 14 | 15 | 13 |
| | P | *0.5852* | ***0.0005*** | *0.0375* | ***0.0348*** | *0.4923* | *0.6773* | *0.8622* | *0.2539* | ***0.0224*** |
| $D_{slow}$ | t-ratio | 0.103 | 1.215 | 1.665 | 0.02233 | 1.423 | 0.3395 | 1.447 | 1.946 | 0.8402 |
| | df | 10 | 18 | 9 | 10 | 13 | 10 | 14 | 15 | 13 |
| | P | *0.9995* | *0.5636* | *0.3421* | *0.9826* | *0.5546* | *0.9331* | *0.5254* | *0.2539* | *0.5320* |
| $F_{slow}$ | t-ratio | 1.055 | 5.524 | 0.2427 | 1.196 | 2.087 | 0.1355 | 3.497 | 1.073 | 2.428 |
| | df | 10 | 18 | 9 | 10 | 13 | 10 | 14 | 15 | 13 |
| | P | *0.7815* | ***0.0002*** | *0.8137* | *0.7768* | *0.3757* | *0.9331* | ***0.0281*** | *0.3180* | *0.0886* |
| $α_{msd}$ | t-ratio | 2.349 | 1.371 | 2.011 | 1.529 | 0.3705 | 2.086 | 2.063 | 5.562 | 4.853 |
| | df | 10 | 18 | 9 | 10 | 13 | 10 | 14 | 15 | 13 |
| | P | *0.2828* | *0.5636* | *0.2895* | *0.6418* | *0.7170* | *0.3256* | *0.3019* | ***0.0004*** | ***0.0022*** |
| $α_{fast}$ | t-ratio | 0.08508 | 4.226 | 2.335 | 0.4044 | 1.533 | 1.054 | 0.7198 | 3.102 | 4.251 |
| | df | 10 | 18 | 9 | 10 | 13 | 10 | 14 | 15 | 13 |
| | P | *0.9995* | ***0.0025*** | *0.2384* | *0.9715* | *0.5546* | *0.6809* | *0.8622* | ***0.0359*** | ***0.0057*** |
| $α_{slow}$ | t-ratio | 1.668 | 0.8014 | 0.7737 | 0.6813 | 1.817 | 1.4 | 2.055 | 1.427 | 1.043 |
| | df | 10 | 18 | 9 | 10 | 13 | 10 | 14 | 15 | 13 |
| | P | *0.5552* | *0.5636* | *0.7073* | *0.9429* | *0.4923* | *0.6549* | *0.3019* | *0.3180* | *0.5320* |
| θ | t-ratio | 0.07485 | 1.243 | 2.091 | 0.1832 | 0.8525 | 2.224 | 0.6676 | 4.093 | 4.164 |
| | df | 10 | 18 | 9 | 10 | 13 | 10 | 14 | 15 | 13 |
| | P | *0.9995* | *0.5636* | *0.2895* | *0.9799* | *0.6512* | *0.3035* | *0.8622* | ***0.0067*** | ***0.0057*** |

Significant results are highlighted in bold. df is degrees of freedom.

identified regions of the PM where spectrin density was highest with more restricted trajectories (Fig. 8 C). Indeed, both the diffusion coefficient and α measured from mean square displacement were reduced, which was also evident specifically in the fast population of molecules (Fig. 8 D and Table 4). The mean turning angle θ was also significantly increased in these regions (Fig. 8 D and Table 4). These observations are consistent with spectrins acting as diffusion barriers, as high densities of filaments would be expected to reduce the number of long-distance paths to displacement (reducing D and α) and cause rebound of PI(4,5)P$_2$:Tubby$_c$ complexes that strike them (increasing θ).

Finally, we interrogated the last component of the cortical cytoskeleton: septins (Fig. 9 A). Septin filaments serve as a scaffold for the F-actin and microtubule cytoskeletons at the PM (Spiliotis and Nakos, 2021), and like spectrins, they are directly associated with the membrane via an interaction with PI(4,5)P$_2$ (Zhang et al., 1999; Tanaka-Takiguchi et al., 2009; Bertin et al.,

2010). We edited a split sfGFP tag into the *SEPT2* allele, yielding the expected 63K septin2–sfGFP complex viewed by in-gel fluorescence (Fig. 9 B). By TIRFM, these edited HeLa cells exhibited localized filamentous patterns on the ventral PM (Fig. 9 B), consistent with a similarly edited cell line from another group (Banko et al., 2019). Gross morphology of septin2–sfGFP was unchanged by Tubby$_c$ expression (Fig. S1 G). Strikingly, we observed that although Tubby$_c$ trajectories crossed into, out of, and through such filaments (e.g., Fig. 9 B), their mobility was greatly altered; overall diffusion by mean square displacement and the diffusion of the fast population were more than halved (Fig. 9 C). Diffusion was also significantly more anomalous (α reduced), and the mean turning angle was significantly increased (Fig. 9 C and Table 4).

The simplest explanation of these data is that, like for spectrin, individual septin filaments present a physical barrier to diffusion. However, since the filaments imaged by

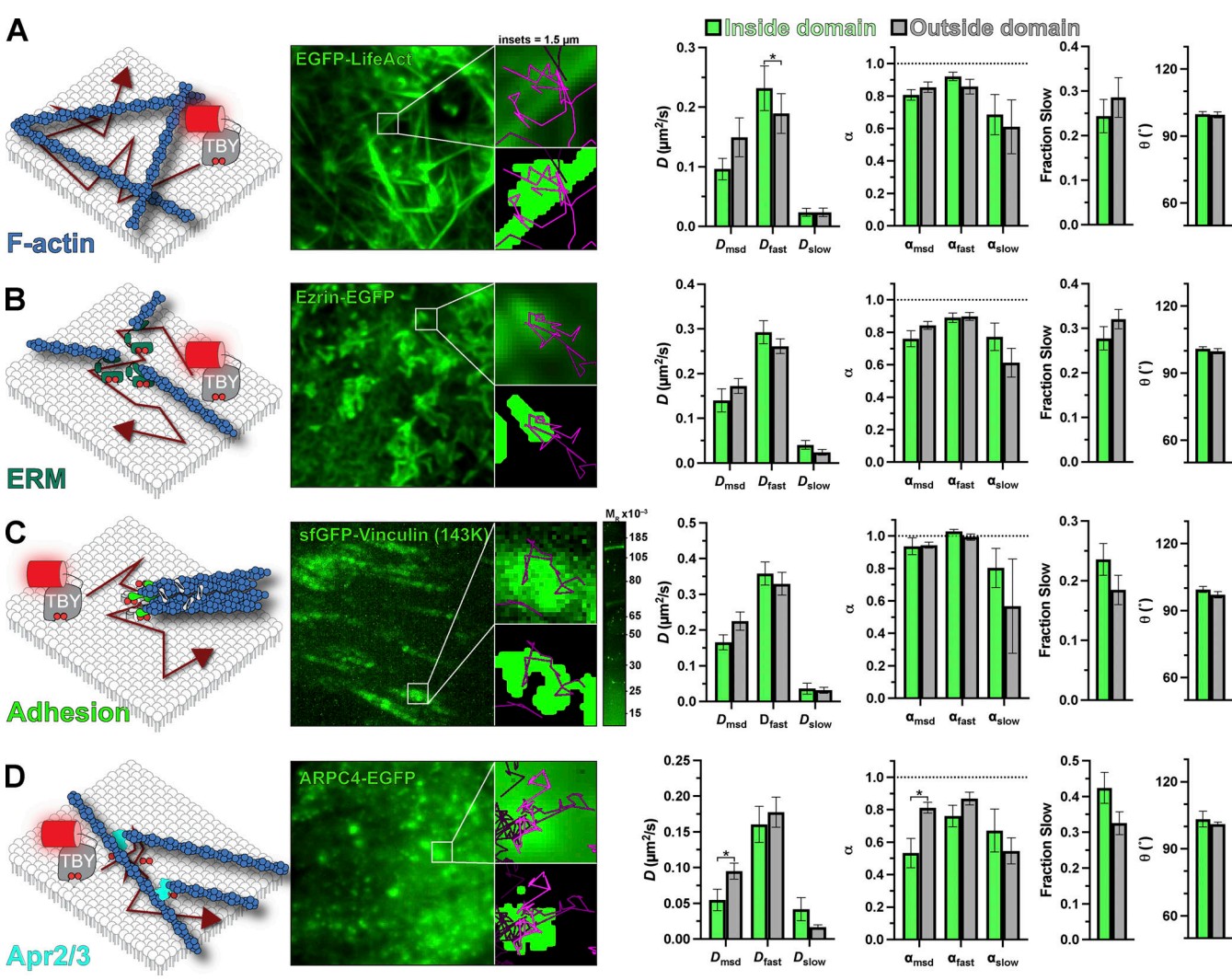

Figure 7.   **The actin cytoskeleton does not present a substantial impediment to PI(4,5)P$_2$ diffusion.** Tubby$_c$-PAmCherry (TBY) single-molecule diffusion was measured inside and outside of the indicated GFP-labeled structure: **(A)** The F-actin cytoskeleton was labeled with LifeAct-GFP. **(B)** Cortical F-actin. Membrane attachment was marked with Ezrin-EGFP expression. **(C)** Focal adhesions were labeled by gene editing the *VCL* locus to express sfGFP-vinculin. **(D)** The Arp2/3 complex was labeled by expression of ARPC4-EGFP. In all cases, the images show the GFP-labeled domain, with the 1.5 µm insets showing an isolated domain as the raw image or thresholded image overlaid with molecule trajectories passing through it during the experiment. Data at right show diffusion coefficients and anomalous diffusion (α) from both mean square displacement analysis of trajectories and analysis of individual displacements. The fraction fast and slow displacements and mean turning angle of trajectories are shown. In all cases, data are the grand means ± SE of 11 (A and C), 14 (B), or 15 (D) cells. The gel image in C shows in-gel fluorescence of sfGFP from lysates of VCL-sfGFP cells, with a single band consistent with the expected M$_r$ of 143,000 for the fusion protein. *P ≤ 0.05 from paired *t* test with Holm–Šidák correction for multiple comparisons.

TIRFM likely represent bundles of many septin filaments, with gaps that cannot be resolved, we sought a more direct test of this hypothesis; we utilized the observation that disruption of the F-actin cytoskeleton causes the septin cytoskeleton to collapse into micron-scale, continuous rings (Xie et al., 1999; Kinoshita et al., 2002). These rings were easily resolved in HeLa cells treated with latrunculin B (Fig. 9 E). Strikingly, trajectories tended to explore the periphery of such rings but very rarely crossed into their center. Quantifying the frequency of such entry into the ring lumens revealed such crossing events happened far less frequently than occurred in the same-sized, random areas of membrane (Fig. 9 F). Therefore, septin2-containing filaments present a bonafide barrier to diffusion of lipids on the

inner leaflet of the PM, as previously suggested (Golebiewska et al., 2011).

## Discussion

Here, we determined the mobility of lipid molecules on the inner leaflet of the PM with ~100 nm and ~100 ms resolution, and we interrogated how this mobility changes in proximity to macromolecular complexes regulating diverse PM functions. Broadly speaking, we find diffusion coefficients of ~0.3 µm²/s (Figs. 2 and 3), agreeing well with prior estimates of PI(4,5)P$_2$ diffusion (Mashanov and Molloy, 2007; Yaradanakul and Hilgemann, 2007; Golebiewska et al., 2008; Hammond et al., 2009) as well as with that of phosphatidylethanolamine in the outer PM leaflet at

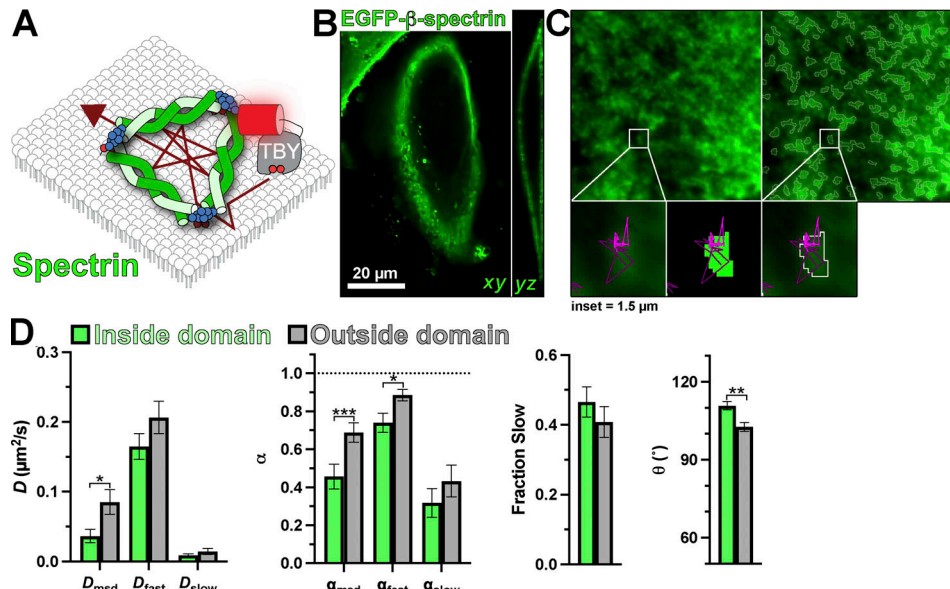

**Figure 8.** **PI(4,5)P₂ diffusion is impaired by the spectrin cytoskeleton. (A)** Schematic of diffusion in the PM proximal to spectrin marked with expressed EGFP-β-spectrin. **(B and C)** Transverse and longitudinal confocal sections (B) and TIRF image (C) of a small PM region of HeLa cells expressing EGFP-β-spectrin (left), with overlaid regions of high spectrin density defined by our local threshold-based segmentation (right); insets show a region of raw and thresholded images overlaid with single-molecule trajectories. Inset = 1.5 μm. **(D)** Diffusion coefficients, α (from msd v Δt plots as well as fast and slow populations), fraction of the population with slow diffusion, and mean turning angles are shown for trajectories classified as inside or outside dense regions of spectrin labeling. Data are the grand means ± SE of 15 cells. *P ≤ 0.05, **P ≤ 0.01, ***P ≤ 0.001 from paired *t* test with Holm–Šidák correction for multiple comparisons.

this level of spatiotemporal resolution (Fujiwara et al., 2002; Andrade et al., 2015). Surprisingly, we found that most PI(4,5)P₂-regulated macromolecular complexes we investigated did not impact the diffusion of PI(4,5)P₂ (Figs. 5, 6, 7, 8, and 9).

One finding that we do not have a completely satisfactory explanation for is the existence of two apparent modes of diffusion in HeLa cells: one rapid and largely Brownian and another slower and more anomalous (Fig. 4). Notably, these two modes are frequently observed in the same trajectory (Fig. 4 A). This observation, when coupled with a slight tendency toward more obtuse turning angles (Fig. 3 C), suggests that the inner leaflet of the PM is largely supportive of Brownian motion, but there is a tendency to produce transient, local confinement. Our data do not allow us to unambiguously assign the molecular cause of this confinement. However, we speculate that high local densities of immobile membrane-anchored proteins, perhaps associated with the cortical cytoskeleton, may transiently trap the lipids long enough to be detected in our experiments. In HeLa cells, the membrane-associated cortical cytoskeleton constrains lipids in 68 nm corrals (Fujiwara et al., 2016). At the spatiotemporal resolution of our experiments, diffusion coefficients, therefore, represent the rate of diffusion between these corrals rather than the more rapid diffusion within them. However, given the variable sizes of such corrals and a stochastic process of escape from them, perhaps our slower population represents lipid molecules trapped within corrals for long enough that the restriction becomes apparent. It is worth mentioning that cortical actin cytoskeletal barriers to lipid diffusion have been shown to be critically dependent on the Arp2/3 complex (Andrade et al., 2015), which is the only complex where we observed a

significant (albeit small) increase in the fraction of displacements exhibiting the slower mode of diffusion (Table 4 and Fig. 4 D).

Whatever the cause of the slower mode of diffusion, it does not seem to be associated with any particular functional complexes probed in this study since the fraction of displacements assigned to this slower mode was not changed substantially in proximity to any of them, except Arp2/3, as discussed above (Figs. 5, 6, 7, 8, and 9; and Table 4). This result surprised us since membrane-attached complexes such as clathrin-coated pits and focal adhesions are envisioned to consist of dense arrays of interlinked, membrane-attached proteins (Schöneberg et al., 2017; Kanchanawong et al., 2010). Even for lipid molecules not directly interacting with these proteins, the dense array of anchored proteins would be expected to block long-distance displacements across the complex—i.e.,- cause percolation of the diffusing lipid. The data presented herein suggest that in fact, the density of membrane-attached proteins is below the percolation threshold for the lipids. Modeling studies have suggested that even with roughly one third of the membrane area occupied by such immobile protein obstacles, impeded diffusion would not be apparent in our spatiotemporal resolution (Saxton, 1994). The implication is that, even though PI(4,5)P₂ itself can be an anchoring component of the membrane proteins, any unbound PI(4,5)P₂ can rapidly and freely diffuse away from the complex. To quote an example, an ∼100 nm clathrin-coated pit would lose an unbound PI(4,5)P₂ molecule from the complex within ∼26 ms, assuming diffusion at 0.3 μm²/s (from r = $\sqrt{4Dt/\pi}$). Therefore, even localized PI(4,5)P₂ synthesis at such structures would rapidly lose any local enrichment unless the newly synthesized lipids were rapidly bound by effectors.

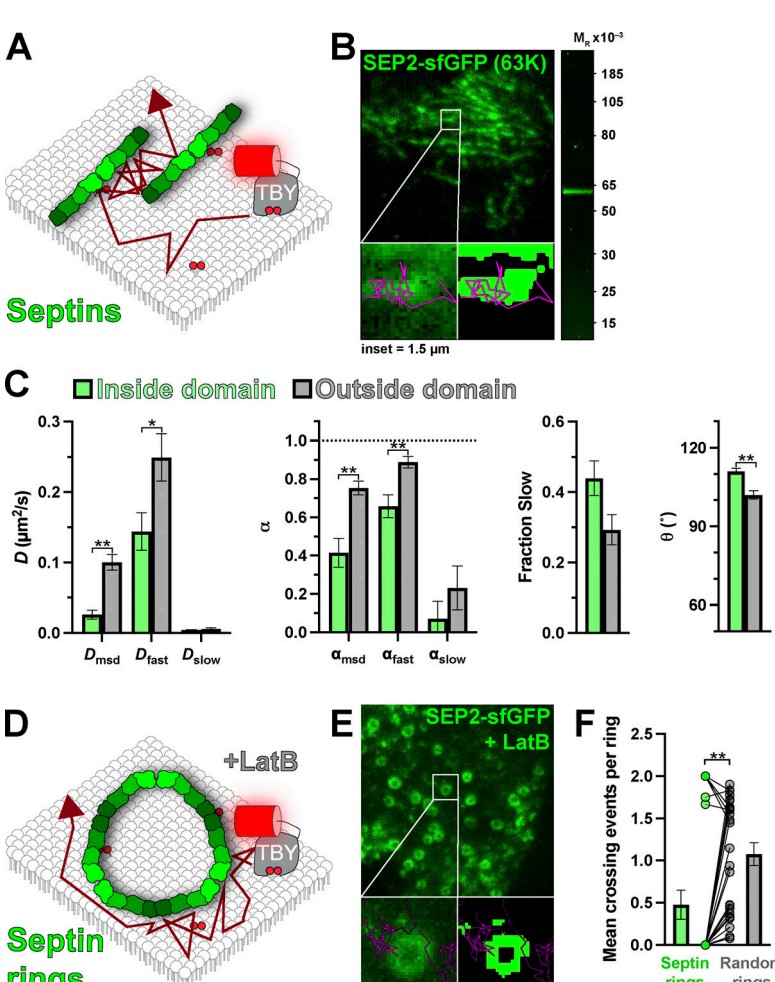

**Figure 9. Septins are a barrier to PI(4,5)P₂ diffusion.**
**(A)** Schematic of diffusion in the PM proximal to septins marked with endogenous septin2 tagged with sfGFP. **(B)** TIRF image of a small PM region of HeLa septin2-sfGFP cell; insets show a region of raw and thresholded images overlaid with single-molecule trajectories. Inset = 1.5 µm. The gel image at right shows in-gel fluorescence of sfGFP from lysates of septin2-sfGFP cells, with a single band consistent with the expected $M_r$ of 63,000 for the fusion protein. **(C)** Diffusion coefficients, α (from msd v Δt plots as well as fast and slow populations), fraction of the population with slow diffusion, and mean turning angles are shown for trajectories classified as inside or outside septin filaments. Data are the grand means ± SE of 14 cells. *P ≤ 0.05, **P ≤ 0.01 from paired $t$ test with Holm–Šidák correction for multiple comparisons. **(D)** Schematic of diffusion around septin rings induced by treatment of cells with latrunculin B. **(E)** TIRF image of a small PM region of latrunculin B-treated HeLa septin2-sfGFP cell; insets show a region of raw and thresholded images overlaid with single-molecule trajectories. Inset = 2 µm. **(F)** Mean number of crossing events into or out of the lumen of septin rings observed in 24 cells is compared with randomized rings (generated by randomizing the masked rings) and shows far fewer crossing events at septin rings. Bars show mean ± SE, points show a pair-wise comparison of individual cells' septin and randomized rings. **P ≤ 0.01 from Wilcoxon matched-pairs signed rank test.

One set of membrane-anchored complexes where we did observe consistent reductions in diffusion was at the spectrin and septin cytoskeletons (Figs. 8 and 9). We could not resolve individual filaments in our TIRFM imaging, so the threshold-defined domains likely represent dense arrays of such filaments. If the filaments are impermeable to lipid diffusion across them, then the convoluted paths between them would explain the slowed and more anomalous diffusion. Direct support for this comes from the induction of resolvable septin rings (Fig. 9 D), which lipid trajectories were very rarely able to cross. It, therefore, seems that septin and spectrin filaments, anchored tightly to the membrane surface by PI(4,5)P₂ molecules (Wang and Shaw, 1995; Zhang et al., 1999; Tanaka-Takiguchi et al., 2009; Bertin et al., 2010), are indeed true barriers to lipid diffusion, including that of free PI(4,5)P₂. In support of this, Langevin simulations have revealed such diffusion barriers for septins (Lee et al., 2014).

The exception presented by spectrin and septin filaments aside, our main conclusion is that the endocytic, cytoskeletal, and organelle tethering complexes that we resolved on the inner leaflet of the plasma membrane do not have the capacity to corral PI(4,5)P₂ or other lipids over the ~0.1 s and ~0.1 µm scales that we resolve—scales that are highly relevant to the assembly and regulation of these complexes.

Rapid PI(4,5)P₂ diffusion in the vicinity of these complexes will quickly dissipate any local enrichment of unbound lipid, even if it is locally synthesized. It, therefore, seems very unlikely that local PI(4,5)P₂ enrichment serves as a platform to induce assembly of components such as clathrin-coated structures or ER–PM contact sites de novo. That is not to say that once these complexes begin to assemble, engagement of their effector proteins with PI(4,5)P₂ will not enrich the lipid and be crucial for growth and regulation of these complexes. Such effector-bound PI(4,5)P₂ is invisible to our lipid biosensors. However, it is implicit from our data that local enrichment of PI(4,5)P₂ must be driven by effector proteins, and not the other way around.

## Materials and methods

### Cell culture and transfection
HeLa (ATCC CCL-2) cells were grown in DMEM (low glucose; 10567022; Life Technologies) supplemented with 10% heat-inactivated fetal bovine serum (10438-034; Life Technologies), 100 U/ml penicillin, 100 µg/ml streptomycin (15140122; Life Technologies), and 1:1,000 chemically defined lipid supplement (11905031; Life Technologies) at 37°C in a humidified atmosphere with 5% $CO_2$.

For transfection, cells were seeded in 35-mm tissue culture dishes with 20 mm number 1.5 cover glass apertures (CellVis) previously coated with 5 µg fibronectin (33016-015; Life Technologies). 1–24 h after seeding, cells were transfected with 0.5 µg of plasmid DNA coding for lipid biosensors. For overexpressed domains, 0.5 µg of plasmid DNA was used. Plasmids were pre-complexed with 3 µg lipofectamine 2000 (11668019; Life Technologies) in 200 µl Opti-MEM (51985091; Life Technologies) according to the manufacturer's instructions. Cells were imaged 4–18 h after transfection. For latrunculin-B treatment (144291; Abcam), cells with Septin2 endogenously labeled with sfGFP were incubated with 1 µM Latrunculin-B for 30 min before imaging.

### Generation of endogenously tagged cell lines

HeLa cells are endogenously tagged with split GFP (Leonetti et al., 2016). HeLa cells stably expressing sfGFP-1-10 (Zewe et al., 2018) were electroporated with 200 pmol single-stranded homologous-directed repair (HDR) template (IDT) and 5 pmol precomplexed gRNA and Platinum Cas9 (Thermo Fisher Scientific) in a 10 µl reaction volume using a Neon Electroporation system (Thermo Fisher Scientific) with a single, 20 ms, 1,500 V pulse according to the manufacturer's instructions. Sequences are provided in Table 5. All HDR templates contained 70-bp homology arms, the GFP-11 sequence, and a flexible linker in frame with the gene to be labeled (5'-CGTGACCACATGGTCCTT CATGAGTATGTAAATGCTGCTGGGATTACAGGTGGCGGC-3'). 48 h after electroporation, cells were sorted by FACS for GFP-positive cells.

### Plasmids

pPAmCherry1-C1 (plasmid 31929; Addgene) was a kind gift of Vladislav Verkhusha (Albert Einstein College of Medicine, New York, NY). The EGFP (*Aequorea Victoria* GFP presenting F64L and S65T mutations) was used to label PM domains. ARPC4 cDNA was obtained from GeneCopoeia. HIV-1-GAG fused with EGFP was purchased from Addgene (plasmid 80605). Ezrin plasmid was a generous gift from Adam Kwiatkowski (University of Pittsburgh School of Medicine, Pittsburgh, PA). All plasmids were verified by dideoxy sequencing. Plasmids were constructed using NEB HiFi assembly or standard restriction cloning. Sources and backbones are indicated in Table 6. Plasmids generated herein will be deposited in Addgene for distribution.

### Microscopy

Cells were imaged in 2 ml FluoroBrite DMEM (A1896702; Life Technologies) supplemented with 25 mM Hepes (pH 7.4) and 1:1,000 chemically defined lipid supplement with 10% heat-inactivated fetal bovine serum on a heated stage at 37°C (Warner instruments). For fixed cell preparations, transfected cells were washed with warmed PBS and immediately fixed with 0.2% glutaraldehyde dissolved in PBS. After 15 min incubation, cells were rinsed three times with freshly diluted 10 mg/ml sodium borohydride dissolved in PBS. Finally, cells were washed twice with PBS and imaged.

All experiments were performed on a Nikon TiE microscope equipped with a TIRF illuminator, a 100× 1.45 NA plan-apochromatic oil-immersion objective, and an Oxxius L4C

Table 5. **HDR and gRNA sequences for targeted genes**

| Gene | gRNA sequences |
|---|---|
| ESYT1 | 5'-TAATACGACTCACTATAGGAATGGAGCGATCTCCAGGAGGTTTAA GAGCTATGCTGGAA-3' |
| CLTA | 5'-TAATACGACTCACTATAGGGCCATGGCGGGCAACTGAAGTTT AAGAGCTATGCTGGAA-3' |
| VCL | 5'-TAATACGACTCACTATAGGCGTATGAAACACTGGCATCGGTTTAA GAGCTATGCTGGAA-3' |
| SEPTIN2 | 5'-TAATACGACTCACTATAGGGCTCTCGGGCACCACGTGTAGTTTAA GAGCTATGCTGGAA-3' |
| ACTN4 | 5'-TAATACGACTCACTATAGGGTATGGCGAGAGCGACCTGTGGTTTA AGAGCTATGCTGGAA-3' |

| | HDR sequences |
|---|---|
| ESYT1 | 5'-ATCGCAAGACTAGGCAACCTCCAGCCAGTCCCTGGGTCGGGC GGATCCTCCCAGAGGTGGCACAATGGAGCGTGACCACATGGTCCT TCATGAGTATGTAAATGCTGCTGGGATTACAGGTGGCGGCCGATC TCCAGGAGAGGGCCCCAGCCCCAGCCCCATGGACCAGCCCTCTGC TCCCTCCGACCCCACTGACC-3' |
| CLTA | 5'-CGGGCGTGGTGTCGGTGGGTCGGTTGGTTTTTGTCTCACCGT TGGTGTCCGTGCCGTTCAGTTGCCCGCCATGCGTGACCACATGGT CCTTCATGAGTATGTAAATGCTGCTGGGATTACAGGTGGCGGCGC TGAGCTGGATCCGTTCGGCGCCCCTGCCGGCGCCCCTGGCGGTCC CGCGCTGGGGAACGGA-3' |
| VCL | 5'-CGCCGGTTCCCGGCCCCGTGGATCCTACTTCTCTGTCGCCCG CGGTTCGCCGCCCCGCTCGCCGCCGCGATGCGTGACCACATGGTC CTTCATGAGTATGTAAATGCTGCTGGGATTACAGGTGGCGGCCCA GTGTTTCATACGCGCACGATCGAGAGCATCCTGGAGCCGGTGGCA CAGCAGATCTCCCACCTGGTGAT-3' |
| SEPTIN2 | 5'-GATGCAGGCGCAGATGCAGATGCAGATGCAGGGCGGGGATGG CGATGGCGGGGCTCTCGGGCACCACGTGGGTGGCGGCCGTGACCA CATGGTCCTTCATGAGTATGTAAATGCTGCTGGGATTACATAAGG TGATGTGCACATATCAAGAAGTCAGAGGTAGGCCCTGTTGTCCCT TAGCCTGGAAGACAGGCAGT-3' |
| ACTN4 | 5'-CCCTGACGCCGTGCCCGGTGCCCTCGACTACAAGTCCTTCTCCAC GGCCTTGTATGGCGAGAGCGACCTGGGTGGCGGCCGTGACCACAT GGTCCTTCATGAGTATGTAAATGCTGCTGGGATTACATGAGGCCC CAGAGACCTGACCCAACACCCCCGACGGCCTCCAGGAGGGGCCTG GGCAGCCCCACAGTCCC-3' |

combiner equipped with 405, 488, 561, and 640 nm lasers. Single-molecule imaging was registered on a Zyla 5.5 sCMOS camera (Andor) with no pixel binning in a rolling shutter mode. Detection of single molecule and domains were recorded with a frame delay of 0.025 ms by using a sequential acquisition between green (488 nm laser line excitation for the EGFP tagged PM domains) and red (561 nm excitation for PAmCherry) on triggering mode controlled by NIS-Elements software. Final exposure for single-molecule imaging resulted in 55 ms by using 30% laser power with 561 nm and from 0.5 to 3 s of 0.8% of 405 nm laser for photoactivation immediately before starting the experiment. The green channel was excited with 3–8% of 488 nm laser. Time-lapse images were recorded in a 16 × 16 µm region of the PM for 30 s.

### Single molecule analysis using thunderstorm

Spot size and brightness (Fig. 1, C and D) were estimated with Fiji thunderstorm plugin (Ovesný et al., 2014). Fixed cells

Pacheco et al.
Free diffusion of PI(4,5)P2 in the plasma membrane

Journal of Cell Biology 13 of 17

**Table 6.  Plasmids used in this study**

| Plasmid | Backbone | Insert | Reference |
|---|---|---|---|
| PAmCherry1-N1-Tubbyc | pPAmCherry-N1 | *Mus musculus* Tub (243-505): PAmCherry1 | This study |
| PAmCherry-N1-FRB-Tubbyc | pPAmCherry-N1 | *Mus musculus* Tub (243-505):MTOR(2021-2113):PAmCherry1 | This study |
| mCherry-N1-Tubby$_c$ | pEGFP-N1 | *Mus musculus* Tub (243-505): mCherry | Quinn et al., 2008 |
| PAmCherry1-N1-PHPLCd1 | pPAmCherry-N1 | PLCD1v2(1-170):PAmCherry1 | This study |
| NES-PAmCherry-cPHx3 | pPAmCherry-C1 | PAmCherry1:PLEKHA1(169-329):GGSGGSGG: PLEKHA1(169-329): GGSGGSGG: PLEKHA1(169-329): | Goulden et al., 2019 |
| PAmCherry1-C1-P4C | pPAmCherry-C1 | PAmCherry1: *L. pneumophila* SidC (608-773) | This study |
| PAmCherry1-C1-P4Mx2 | pPAmCherry-C1 | PAmCherry1: *L. pneumophila* SidM(546-647):SidM(546-647) | This study |
| PAmCherry1-C1-LACT-C2 | pmEos2-C1 | PAmCherry1: *Bos taurus* MFGE8 (274-431) | This study |
| PAmCherry1-N1-Lyn11 | pmEos2-N1 | LYN(1-11):PAmCherry1 | This study |
| ARPC4 | pReceiver-M98 | EGFP:ARPC4 | GeneCopoeia |
| EGFP-N-Ezrin | pmCherry-N1 | *Mus musculus* EZR(1-586):EGFP | This study |
| EGFP-C3-β-spectrin | pEGFP-C3 | EGFP:SPTB(1-2364) | This study |
| EGFP-N1-Lifeact | pEGFP-N1 | LifeAct:EGFP | Tamas Balla |
| EGFP-C1-FKBP-CYB5$^{tail}$ | EGFP-C1 | EGFP:FKBP:[GGSA]$_4$GG:CYB5A(100-134) | This study |

expressing PAmCherry1-N1-Tubby$_c$ were imaged using the same microscopic settings as for live cell single-molecule time lapses. Then, raw images were run in the thunderstorm plugin. Settings for localization of molecules were determined by using a wavelet filter with a local maximum method and an integrated Gaussian Point spread function.

For fluorescence intensity per spot, histograms of photon counts were generated with a bin size of five photons. For size of spots, histograms of 2 SD of detection (Sigma-Aldrich) were produced with a bin size of 4 nm.

### Analysis of the trajectories

Single-molecule trajectories were generated using the open-source Fiji TrackMate (Tinevez et al., 2017). For single-molecule detection, a difference of Gaussians filter was used with an estimated diameter of 0.5 μm (i.e., an ~8 × 8 pixel neighborhood). For trajectory assembly, an implementation of a simple linear assignment problem algorithm with 0.7 μm maximal distance for both linking and gap-closing was used. Coordinates of trajectories were exported as CSV files and analyzed using custom-written code in Python to calculate trajectory displacements. For all analyses, only those trajectories longer or equal to 0.88 s (16 frames) were used. This time frame was selected as a minimum to allow sufficient time lags for mean square displacement analysis of individual trajectories, including a robust analysis of α, as described next. For consistency's sake, the same filtered data were used for that analysis and the analysis of independent displacements.

$$MSD_{1-16} = 4Dt^\alpha. \quad (1)$$

Individual trajectories were analyzed through the mean square displacement (*MSD*; Vrljic et al., 2007). The trajectories were cut at a duration of 0.88 s and the *MSD* was calculated from the first 15 time lags. The diffusion coefficient was extracted from a power of law function using the 16 time lags of *MSD* plots.

All independent displacements (r) from all trajectories were pooled to construct cumulative distribution functions at different time lags (iΔt; Vrljic et al., 2002). The cumulative distribution function plots were constructed for the first four time lags. Unless otherwise mentioned, each curve was fit with Eq. 2, which considers two populations of diffusion coefficients:

$$P(r, i\Delta t) = 1 - \left( exp\left[\frac{(-r^2)}{4D_{fast}(i\Delta t)}\right] + exp\left[\frac{(-r^2)}{4D_{slow}(i\Delta t)}\right] \right), \quad (2)$$

where $D_{fast}$ and $D_{slow}$ are the diffusion coefficients for the fast and the slow population, respectively. $D_{fast} > D_{slow}$ at all the time lags. To analyze whether diffusion is Brownian or anomalous, the change of diffusion coefficient over time was evaluated for each population:

$$D = D_n t^{\alpha-1}, \quad (3)$$

where the exponent α denotes the grade of anomalous motion. $D_n$ is either $D_{fast}$ or $D_{slow}$. For Brownian motion, the exponent α is close to 1. For subdiffusive and superdiffusive motion, α is <1 or >1, respectively. Fitting was performed using Graphpad Prism 9.

### Analysis of diffusion in-domains

The trajectories were obtained through sequential acquisition (delayed time of ~27 ms) between PM domains (green channel) and single molecule detection (red channel). The expression of

EGFP-labeled domains was used to generate binary masks through à trous wavelet decomposition (Olivo-Marin, 2002), as described previously (Hammond et al., 2014).

To quantify diffusion coefficients inside domains, we used a moving window approach. A moving window was defined as a unit of analysis of a partial segment of trajectory with a length of five consecutive localizations ($i\Delta t = 5$). The moving window is scanned through the trajectories step by step until the whole trajectory is covered. All those partial trajectories produced by the moving windows were categorized as an in-domain measurement if at least one localization fell inside of a domain. In the same way, all moving windows in which all localizations were outside of domains correspond to out-of-domain measurement. Each partial segment was used to construct cumulative distribution functions of radial displacements (Eq. 2). Diffusion coefficients for fast and slow populations were calculated using the first four time lags (Eq. 3). Additionally, all partial trajectories produced by the moving windows were analyzed by MSD (Eq. 1), and the median was graphed per cell.

Analysis of septin rings was performed exclusively in those domains that under the binary mask were completely closed rings. From trajectories, the number of crosses was counted. A crossing was defined as a segment produced by the Euclidean vector between localizations that move through the inner side of the ring or vice versa. For control, the crosses were counted in randomized binary masks produced by flipping vertically, horizontally, and vertical-horizontally the same masks with septin rings produced from experimental data.

### Turning angles

The turning angles were quantified from the resultant angle between two consecutive vectors along the trajectories. Only trajectories longer than 16 frames were used. To measure the turning angles in domains, only whole trajectories moving through domains (same binary masks from previous section) were considered for the analysis. Here, an in-domain measurement was established if at least one localization in a vector is positioned in the coordinates corresponding to a domain. An out-of-domain measurement was all the turning angles from segments of vectors moving out of domains.

### Statistical analysis

All statistical analyses were performed using Graphpad Prism 9. Trajectories were analyzed as above, and the mean values were computed based on the total trajectories recorded from individual cells. Note that data were collected from ≥3 independent experiments, though variability among cells in each experiment was greater among cells in a given experiment than it was between experiments. Thus, we define the cell as the unit of biological variability. In the data collected from individual domains (reported in Figs. 4, 5, 6, 7, and 8), variability in the number of trajectories interacting with domains led to a small subset of cells with highly divergent estimates of diffusion coefficients and α for the slow population, which skewed the mean. Therefore, we subjected all these data to outlier analysis using the robust regression followed by outlier identification method (Motulsky and Brown, 2006), setting a maximum false discovery rate (Q) of 0.1%. Any cell with a detected outlier in any parameter was excluded from further analysis. Statistical tests were performed as described in the figure legends for each experiment and detailed in Tables 1, 2, 3, and 4. Data were subject to the D'Agostino and Pearson normality test, which revealed parametric distributions, with the exception of the data presented in Fig. 9 F.

### Online supplemental material

Fig. S1 shows that Tubby$_c$-mCherry expression does not alter PM domain architecture.

## Acknowledgments

The authors are grateful to Scott Hansen (University of Oregon, Eugene, OR) for his critical reading of the manuscript and helpful suggestions.

This work was supported by National Institutes of Health grant 2R35GM119412.

The authors declare no competing financial interests.

Author contributions: J. Pacheco, A.C. Cassidy, and G.R.V. Hammond performed single molecule tracking experiments (investigation) and analyzed data (formal analysis). J. Pacheco, J.P. Zewe, and R.C. Wills prepared gene-edited cell lines (methodology). G.R.V. Hammond acquired funding and wrote the original draft. All authors reviewed and edited the manuscript.

Submitted: 26 April 2022

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

# Supplemental material

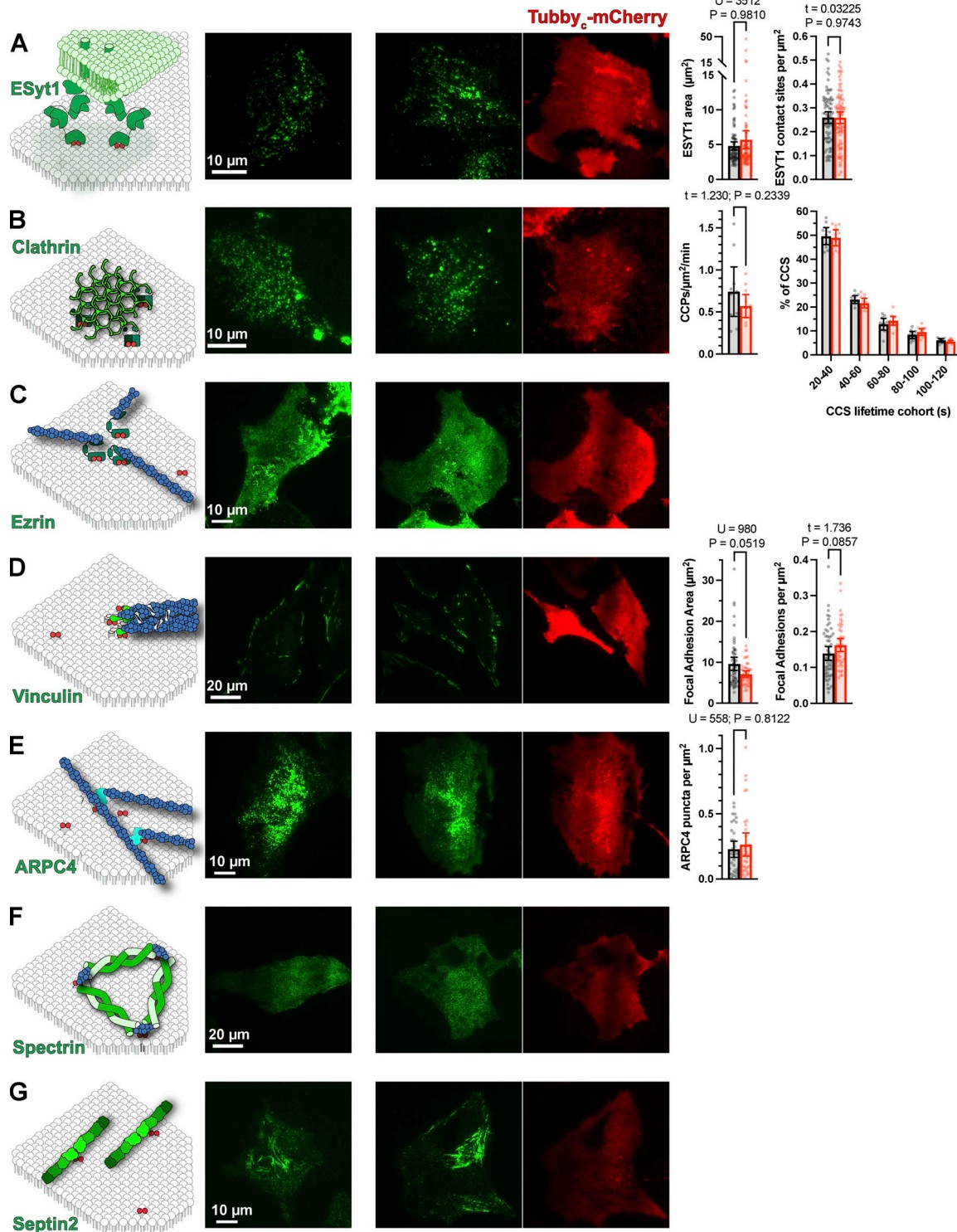

Figure S1.   **Tubby_c-mCherry expression does not alter PM domain architecture. (A)** examples of sfGFP-E-Syt1 edited cells, and the mean area and density of contact sites for 80 (control) or 88 (Tubby_c-mCherry) cells were measured. P values and statistics are given for the results of a Mann–Whitney U-test or Unpaired t test as indicated. Bars show mean ± 95% C.I. **(B)** Examples of sfGFP-CLTA edited cells, with the mean clathrin-coated structure initiation rate and lifetime cohorts of 10 (control) or 11 Tubby_c-mCherry expressing cells. Results of an unpaired t test are indicated. Bars are mean ± 95% C.I. **(C)** Representative examples of Ezrin-EGFP–expressing cells with and without Tubby_c-mCherry. **(D)** Examples of sfGFP-vinculin edited cells; mean vinculin-labeled adhesion area and density was measured in 55 (control) or 46 (Tubby_c-mCherry expressing) cells. P values and statistics are given for the results of a Mann–Whitney U-test or Unpaired t test as indicated. Bars show means ± 95% C.I. **(E)** Images of ARPC4-EGFP expressing cells with and without Tubby_c-mCherry. The density of ARPC4-labeled puncta were quantified in 34 cells from each group; P values and U value are given for the results of a Mann–Whitney U-test. **(F and G)** Representative images of EGFP-β-spectrin expressing (F) or Septin2-sfGFP edited (G) cells, with or without Tubby_c-mCherry expression. Throughout, data were subject to a D'Agostino & Pearson normality tests to determine whether t or U-tests were performed.

