## [Peer Review File · The Journal of Cell Biology]

PI(4,5)P2 Diffuses Freely in the Plasma Membrane Even Within High Density Effector Protein Complexes

Jonathan Pacheco, Anna Cassidy, James Zewe, Rachel Wills, and Gerry Hammond

Corresponding Author(s): Gerry Hammond, University of Pittsburgh School of Medicine

Review Timeline:

Submission Date:	2022-04-26
Editorial Decision:	2022-05-12
Revision Received:	2022-08-11
Editorial Decision:	2022-09-19
Revision Received:	2022-10-21

Monitoring Editor: William Prinz

Scientific Editor: Lucia Morgado-Palacin

Transaction Report:

DOI: <https://doi.org/10.1083/jcb.202204099>

Revision 0

Review #1

1. Evidence, reproducibility and clarity:

Evidence, reproducibility and clarity (Required)

Pacheco, et al combine live cell imaging and photoactivated single particle tracking (sptPALM) to measure spatial and temporal features PIP2 diffusion in the plasma membrane of HeLa cells. PIP2 binding domains are expressed and visualized in TIRF. Additionally, co-expression with markers for plasma membrane domains or effectors permitted measurement of the impact of these features on PIP2 mobilities. In general, the work provides an argument that pooling of unbound PIP2 is not a likely mechanism for enhanced effector binding, but, rather, that effectors may be enriching PIP2 at the membrane. The work is clearly presented and demonstrates substantial effort but fails to provide evidence that normal cellular functions are preserved in the context of the measurements.

Major comments

1. The biggest concern in interpreting the data is results from the competitive binding of the lipid biosensors. Particularly in the context of measurements with co-expressed constructs, normal kinetics or measurements of cellular processes should be included. The simplest example is the kinetic of CME. Given the known dependence on PIP2, any perturbation in this kinetic confounds the interpretation of the data substantially.
2. All of the data are acquired in cells that have been constantly maintained in complete media. What are the effects of starvation and stimulation on PIP2 behaviors as read out by these sensors?
3. For diffusional analyses, only trajectories longer than 0.88 s (16 frames) were used. This is never justified. All data could be fit using a step size distribution and may be informative, especially given the observations of two mobility populations.
4. Masking of spectrin data (Fig. 8), are not very compelling. A much more complete presentation of the data are required, especially to justify the strong conclusions.

Minor comments

1. In Figure 2C, there is evidence of bimodality in the diameter distribution. This should be discussed.
2. In the absence of direct visualization of preexisting PIP2-effector complexes, some immunofluorescence (or similar approach) would be useful to strengthen the conclusions regarding the order of PIP2 vs effector accumulation at the surface, especially since this is the most bold claim in the work.

2. Significance:

Significance (Required)

This work could be important in the growing field of PIP-mediated regulation of the membrane and signaling processes. This work is applicable to those measuring PIP-mediated recruitment of signaling macromolecules (i.e. PLC) as well as the polarization communities. Given my expertise in single particle tracking and cellular decision-making, the methodologies are effectively justified, including shortcomings.

****Referees cross commenting****

Of the 2/3 reviews shared, there is a consensus that there is insufficient characterization of the behavior of the biosensors, especially to point to freely diffusing pools. This seems to be the major drawback that could substantially confound the data interpretation.

3. How much time do you estimate the authors will need to complete the suggested revisions:

Estimated time to Complete Revisions (Required)

(Decision Recommendation)

Between 3 and 6 months

4. Review Commons values the work of reviewers and encourages them to get credit for their work. Select 'Yes' below to register your reviewing activity at Publons; note that the content of your review will not be visible on Publons.

Reviewer Publons

Yes

Review #2

1. Evidence, reproducibility and clarity:

Evidence, reproducibility and clarity (Required)

In this paper, Pacheco and colleagues investigated the diffusion of different biosensors of PI(4,5)P₂, PI4P, and PS over the membrane using single particle tracking photoactivation localization microscopy. The authors also performed an in-depth analysis to find out whether PI(4,5)P₂ diffusion is altered in a series of cellular structures which are usually considered PI(4,5)P₂ dependent, such as the ER:PM contact sites, clathrin-coated endocytotic cups, actin crosslinkers, and other components of cortical cytoskeleton. The authors concluded that apart from spectrin and septin based cytoskeleton structures, the PI(4,5)P₂ diffusion remain practically unchanged compared to the bulk which may indicate that the those complex cellular structure formations are not organized by an enriched level of PI(4,5)P₂.

Major Comments:

1. The authors did not explain why they are considering that only "free" lipids are involved in their diffusion studies and why the population of "bound" lipids must be distinct from the "free" counterpart? Intuitively, when new PI(4,5)P₂ molecules are synthesized on the membrane they can either bind to effectors or biosensors and also possibly dynamically interchange with both. In fact, as authors are possibly aware, in some cases, PH domains were shown to be hinder agonist induced hydrolysis of PI(4,5)P₂ (Varnai, Balla JCB 1998) which suggests the pools of PI(4,5)P₂ may be similar.
2. The authors used different biosensor diffusion as a proxy of lipid diffusion. While that is a major established way of visualizing lipid diffusion, the authors need to explain how they accounted for different physiological processes that would cause the biosensors to be lost from the membrane? For example, as it was shown in different systems, membrane bound biosensors fall off to cytosol when a lipid is degraded. Additionally, biosensors often turnover very fast. For e.g. at single molecule level, PIP₃ biosensor was shown to bind to PIP₃ for only ~120 ms (Matsuoka et al. J Cell Sci 2006). Did the authors observe such fast on and off mechanism (shuttling) with any of biosensors in their system and if they did, how did they update their diffusion analysis?
3. Why the authors preferred Tubbyc over PH-PLC δ for studies in complex structures is unclear. The argument of "Tubbyc behaves most similarly to the other lipid sensors" is weak as a PI(4,5)P₂ sensor does not necessarily need to behave similarly to a PI4P/PS sensor to be a better sensor. It would be interesting to see PH-PLC δ data in some of these structures.
4. Since the main claim of the paper is the unhindered PIP₂ diffusion in the complex structures, it would be a great addition if, as a proof concept, at least one of the structures can be imaged in super resolution mode while doing the single molecule tracking in other channel, as that would show the finer details of the structures and demonstrate that the results were valid.
5. What is the molar ratio of expression level of Spectrin/Septin vs others like CLC/Vinculin/ESyt1? Is it possible that if the second set of molecules are expressed in equivalent amounts, those can also hinder the PI(4,5)P₂ diffusion?

Minor Comments:

1. PI(4,5)P₂ is mentioned as "master regulator" in multiple places in the paper (e.g. line 22, 50), whereas the current understanding in the signaling and cytoskeletal research field is that no

particular protein or lipid can be considered as the "master regulator" due to the existence of multiple redundancies, crosstalks, and feedback loops.

2. Quite a few images (and their insets) do not have scale bars.

3. The numbering in Figure 4 is not correct - the figure referred as figure 4C in line 193 and 194 actually remained unnumbered in figure.

4. The authors should delineate in the Discussion how different the lipid enrichment would be if PI(4,5)P2 are locally synthesized in those special structures, as has been claimed (for e.g. Jung S-R et al. PNAS 2021 showed it in clathrin coated pits).

2. Significance:

Significance (Required)

While many lipid biologists have been studying the membrane PI(4,5)P2 diffusion for more than a decade using different biosensors and fluorescent analogs, this study does present a rigorous report on a set of complex experiments to find the diffusion coefficients of different lipid biosensors associated with various specialized cellular structures and thus should still be interesting to phosphoinositide biologists (provided the authors satisfactorily addresses the concerns). The main claim in the paper that the "free" PI(4,5)P2 is not enriched in those specialized structure is novel. In our opinion, however, the conceptual advance is compromised by the caveats explained above. The paper did not demonstrate its physiological significance in terms of the formation/activity of those specialized cellular structures, especially when "bound" PI(4,5)P2, according to their model, still get enriched, and thus can help formation of the spatialized structure by a positive feedback.

****Referees cross commenting****

We could read only Reviewer #1's comment. Overall, most of the points of raised by the Reviewer #1 are valid concerns. Although, we believe that repeating these experiments in starvation condition (Major comment #2) is kind of outside the scope of the present work. The authors should try to address all other concerns in full before publication in an appropriate journal.

3. How much time do you estimate the authors will need to complete the suggested revisions:

Estimated time to Complete Revisions (Required)

(Decision Recommendation)

Between 1 and 3 months

4. Review Commons values the work of reviewers and encourages them to get credit for their work. Select 'Yes' below to register your reviewing activity at Publons; note that the content of your review will not be visible on Publons.

Reviewer Publons

Yes

Revision Plan

Manuscript number: RC-2022-01237

Corresponding author(s): Hammond, Gerald

1. General Statements [optional]

Our paper reveals rapid and free diffusion of phosphoinositide lipids even in regions of the plasma membrane where high-density effector protein complexes assemble. This has fundamental implications for how lipid regulate proteins. We were heartened to see that both reviewers appreciated the novel findings, especially given that, as we describe below in detail, we are able to fully address all of their criticisms.

A note shared by both reviewers, articulated in their cross-commenting and assessment of the manuscript's significance, is that the limitations of using biosensors to measure lipid mobility leaves open the possibility of effector-bound lipid that still produce localized signaling effects, and that this possibility diminishes the conceptual advance of the manuscript. Our article acknowledges this possibility, which really doesn't diminish the conceptual advance at all; our last sentence is key here: *"it is implicit from our data that local enrichment of PI(4,5)P₂ must be driven by effector proteins, and not the other way around."* This concept is lost in most of the current literature, which conflates the idea of localized lipid enrichment with the ability to direct protein complex assembly *a priori*. Papers that have advanced this questionable idea (lipid enrichment before lipid-effector binding) have been "high impact": *Nature*, *PNAS*, *J Cell Biol*, *Science* to quote a few we cite herein. On the other hand, papers that argue against localized lipid pools are relegated to "lower impact" journals: e.g. *J Biol Chem* and *Traffic*. There is therefore somewhat of an "excitement bias" in the literature, where the higher impact papers on one side of the argument skew the weight of the evidence unfairly. We humbly invite you, as the editor of a high impact journal, to help redress this imbalance in the literature by publishing our revised manuscript.

In our revision plan, points are numbered X.Y, where X is reviewer and Y is point number (with minor points numbered sequentially after their major points).

2. Description of the planned revisions

1.1 - *The biggest concern in interpreting the data is results from the competitive binding of the lipid biosensors. Particularly in the context of measurements with co-expressed constructs, normal kinetics or measurements of cellular processes should be included. The simplest example is the kinetic of CME. Given the known dependence on PIP₂, any perturbation in this kinetic confounds the interpretation of the data substantially.*

As we describe in the opening paragraph of the results, we believe that the biosensors, especially at the low levels they are expressed at for single molecule imagine, are unlikely to perturb physiologic function because such a small fraction of lipid is sequestered. Nonetheless, we agree with the reviewer that the manuscript is substantially strengthened by an empirical

Revision Plan

demonstration of this fact. We therefore propose to include the following experiments, where the experimental parameters will be tested in mock vs Tubby_c-mCherry expressing cells:

Domain	Experiment
E-Syt1 (ER:PM MCS)	E-Syt-1 puncta size and density
Clathrin light chain (clathrin coated structures)	Clathrin coated structure initiation rate and lifetime distribution.
Ezrin-EGFP (F-actin membrane binding)	Ezrin-EGFP morphology
sfGFP-Vinculin (focal adhesions)	sfGFP-vinculin-marked focal adhesion size and density
ARPC4-EGFP (Arp2/3)	ARPC4-EGFP puncta size and density
EGFP-β-spectrin (spectrin cytoskeleton)	EGFP-β-spectrin morphology
sfGFP-Septin2 (septin cytoskeleton)	sfGFP-Septin2 morphology

We will thus be able to demonstrate that these membrane structures are unperturbed by biosensor expression.

1.4 - *Masking of spectrin data (Fig. 8), are not very compelling. A much more complete presentation of the data are required, especially to justify the strong conclusions.*

We agree with the reviewer that these data do not enable the reader to fully scrutinize the distribution of spectrin, and how diffusion changes at high sites of high spectrin density. In the revised manuscript, we will include the complete area of thresholded membrane (with an inset to show a representative overlaid trajectory).

2.3 - *Why the authors preferred Tubby_c over PH-PLCδ for studies in complex structures is unclear. The argument of "Tubby_c behaves most similarly to the other lipid sensors" is weak as a PI(4,5)P₂ sensor does not necessarily need to behave similarly to a PI4P/PS sensor to be a better sensor. It would be interesting to see PH-PLCδ data in some of these structures.*

Firstly, we have amended the first paragraph of this section of the manuscript (**p. 9, line 235**) to clarify the selection of Tubby_c over PH-PLCδ1: *"For these experiments, we elected not to employ the PH-PLCδ1, since we found in the previous section that it diffused significantly slower than other lipid biosensors, which can be ascribed to lipid-extrinsic protein-protein interactions (Hammond et al., 2009). We instead employed the Tubby_c PI(4,5)P₂ biosensor, since Tubby_c behaves most similarly to the other lipid sensors (figures 3, 4 and table 3). This is a good indication of unimpeded diffusion of the lipid:biosensor complex, since a lipid-selective biosensor interacts with the headgroup of the lipid, preventing lipid-selective interactions with other proteins that could impair diffusion for the duration of the biosensor complex. Diffusion should therefore be mainly limited by the viscous drag of the acyl chains in the membrane, which is not expected to differ significantly among different lipid classes (or the dually acylated Lyn₁₁ peptide). For Tubby_c, but not PH-PLCδ1, this seems to be the case."*

We therefore are of the opinion that including PH-PLCδ1 is potentially misleading, and best omitted. That said, we include the rebuttal here under "things we will do" because we did collect data with this probe, with the same broad findings but much lower diffusion coefficients. We can include this analysis as a supplement if the editors and reviewers deem it essential to the manuscript.

2.4 - *Since the main claim of the paper is the unhindered PIP2 diffusion in the complex structures, it would be a great addition if, as a proof concept, at least one of the structures can be imaged in super resolution mode while doing the single molecule tracking in other channel, as that would show the finer details of the structures and demonstrate that the results were valid.*

This is a great experiment, though extremely technically challenging. The only super-resolution technique compatible with the high frame rates required for concomitant single particle tracking is photoactivated localization microscopy (PALM) – indeed, sptPALM is what we use for the tracking. However, obtaining adequate localizations to sample a structure at super resolution typically requires the collection of many millions of individual molecular localizations over a much longer time-frame than typically accomplished in our ~minute long spt tracking experiments. We are willing to try these experiments, which are currently underway. However, we cannot guarantee that sufficiently high-quality, high resolution structures will be attainable in parallel with tracking. The likely caveats are that we will not obtain sufficient localizations in a short time frame (~minutes) to (a) produce a super-resolution reconstruction of the structure and (b) that this can occur fast enough relative to the natural dynamics of these structures – e.g. a clathrin coated pit that turns over in < 2 min. We therefore ask that the decision to accept the manuscript not be hinged on success of these experiments.

3. Description of the revisions that have already been incorporated in the transferred manuscript

1.3 - *For diffusional analyses, only trajectories longer than 0.88 s (16 frames) were used. This is never justified. All data could be fit using a step size distribution and may be informative, especially given the observations of two mobility populations.*

We thank the reviewer for spotting this omission. We have now amended the methods on **p 19, line 525** with the requested information: *“For all analyses, only those trajectories longer or equal to 0.88 s (16 frames) were used. This time frame was selected as a minimum to allow sufficient time lags for mean square displacement analysis of individual trajectories, including a robust analysis of alpha, as described next. For consistency’s sake, the same filtered data were used for that analysis and for the analysis of independent displacements.”*

The reviewer is correct, we could have used all displacements for the step size distribution (which we performed by cumulative step size distribution), but we did not for consistency with the individual trajectory data. However, an analysis for Tubby_c comparing the data presented in figure 3 shows that the same result is obtained when the filtered data (≥ 0.88 s trajectory length) are used as when all data (i.e., ≥ 2 localizations in subsequent frames, separated by 55 ms) are used:

Revision Plan

1.5 - In Figure 2C, there is evidence of bimodality in the diameter distribution. This should be discussed.

The reviewer has a keen eye! We have now added an explanation of this observation in the results on **p 5, line 129**: “We did note a minor peak of fluorescence spot size centered around the 65-75 nm bin (**figure 2C**), representing less than 2.5% of all localizations. Given our pixel size of 65 nm in image space, we believe this is caused by “hot pixels” in our sCMOS that locally enhance signal and lead to an artifactually narrower distribution in a few single molecule localizations.”

2.1 - The authors did not explain why they are considering that only “free” lipids are involved in their diffusion studies and why the population of “bound” lipids must be distinct from the “free” counterpart? Intuitively, when new PI(4,5)P2 molecules are synthesized on the membrane they can either bind to effectors or biosensors and also possibly dynamically interchange with both. In fact, as authors are possibly aware, in some cases, PH domains were shown to be hinder agonist induced hydrolysis of PI(4,5)P2 (Varnai, Balla JCB 1998) which suggests the pools of PI(4,5)P2 may be similar.

We completely agree with the reviewer here, who has clearly articulated a key point of our interpretation that we failed to communicate in the opening paragraph of the results section and figure 1. Our mistake was to use the word “free” for a lipid, which has two meanings in our manuscript: (i) unhindered, rapid and Brownian diffusion of lipids and (ii) unbound lipids. We have now clarified by using the term “unbound” where we meant it, instead of “free”. Hopefully,

the revised figure 1 and opening paragraph of the results on **p. 4, line 92** clarifies sufficiently to address the reviewer's concern:

*“PI(4,5)P₂ in the PM exists in a rapid dynamic equilibrium with its effector proteins (**figure 1**). Estimates using fluorescent acyl chain derivatives indicate that two out of three PI(4,5)P₂*

*molecules are in complex with such proteins at any given moment (Golebiewska et al., 2008). In this manuscript, we consider the remaining one third of lipid molecules that are estimated to be unbound. When new complexes are assembled, or new effector proteins are recruited to these complexes, it is this unbound lipid that recruits them; this is the lipid pool that must be locally concentrated to modulate an effector complex. We therefore measured the diffusion of these unbound lipid molecules. To this end, we used genetically encoded lipid biosensors, which interact with the headgroup. The biosensors themselves are in rapid dynamic equilibrium with the lipids and preclude interaction with endogenous effector proteins (**figure 1**). Unlike effector proteins, the biosensors are estimated to sequester a much smaller fraction of PI(4,5)P₂, likely less than 10% (Wills et al., 2018). Although expression of high concentrations of biosensor can sequester a higher fraction of the unbound pool, reducing the pool*

available for effector interaction and inhibiting phospholipase C, for example (Várnai and Balla, 1998), this is unlikely to occur in the experiments described below, since low expression levels were utilized to favor resolution of single molecules. Crucially, previous studies have shown that the diffusion coefficient of biosensor-bound PI(4,5)P₂ is unchanged from the unbound lipid (Mashanov and Molloy, 2007; Yaranakul and Hilgemann, 2007; Golebiewska et al., 2008; Hammond et al., 2009).”

2.2 - *The authors used different biosensor diffusion as a proxy of lipid diffusion. While that is a major established way of visualizing lipid diffusion, the authors need to explain how they accounted for different physiological processes that would cause the biosensors to be lost from the membrane? For example, as it was shown in different systems, membrane bound biosensors fall off to cytosol when a lipid is degraded. Additionally, biosensors often turnover very fast. For e.g. at single molecule level, PIP3 biosensor was shown to bind to PIP3 for only ~120 ms (Matsuoka et al. J Cell Sci 2006). Did the authors observe such fast on and off mechanism (shuttling) with any of biosensors in their system and if they did, how did they update their diffusion analysis?*

We have now accounted for the effects of shuttling of the biosensors and enzymatic degradation of the lipid in a new paragraph in the results on **p. 6 line 134**: *“The single biosensor molecules represented complexes with lipids on the membranes imaged in TIRFM. They are readily discerned from unbound, cytoplasmic biosensors diffusing just above but in the plane of the membrane by the relative diffusion rates: lipids diffuse up to ~1 μm²/s in cells, whereas*

*cytosolic biosensors diffuse around $20 \mu\text{m}^2/\text{s}$ (Hammond et al., 2009). Displacement of such molecules in the camera exposure time, $t = 55 \text{ ms}$, can be estimated from $\sqrt{4Dt/\pi}$ (Teruel and Meyer, 2000) at approximately 260 nm and $1.2 \mu\text{m}$ respectively. The large degree of displacement in the latter case “blurs” and spreads the intensity of the single molecule image to the extent that it is not longer resolvable against camera noise. It follows that biosensors can only be tracked for the lifetime that they stay in complex with the lipid. Estimates of the lifetime of Tubby_c molecules extrapolated back to zero laser-induced photobleaching place this binding lifetime at 339 ms (**figure 2E**). Enzymatic turnover of the lipids, or engagement with an endogenous effector protein requires the dissociation of the biosensor first, so these processes do not impact our diffusion measurements. Indeed, we have previously demonstrated that rates of biosensor-bound lipid catabolism are limited by the dissociate rate of the biosensor (Hammond et al., 2009).”*

2.6 - *PI(4,5)P₂ is mentioned as “master regulator” in multiple places in the paper (e.g. line 22, 50), whereas the current understanding in the signaling and cytoskeletal research field is that no particular protein or lipid can be considered as the “master regulator” due to the existence of multiple redundancies, crosstalks, and feedback loops.*

We removed the term “master” from line 50 of the introduction, since it was redundant. We have left the term “a master regulator”, since this is different from “master regulator” or “the master regulator” as implied by the reviewer. In our context, we mean master regulator to mean shared, not necessarily the hierarchically most important. Essentially, like a master key that is able unlock most PM processes, though it is not necessarily the only point of regulation.

2.7 - *Quite a few images (and their insets) do not have scale bars.*

We have amended the figures so that they all include an indication of the inset size in the figure and legend; for many images, the inset box also serves as the scale information

2.8 - *The numbering in Figure 4 is not correct - the figure referred as figure 4C in line 193 and 194 actually remained unnumbered in figure.*

We thank the reviewer for spotting this error on our part. This is now fixed on **p. 8 lines 217-226**.

2.9 - *The authors should delineate in the Discussion how different the lipid enrichment would be if PI(4,5)P₂ are locally synthesized in those special structures, as has been claimed (for e.g. Jung S-R et al. PNAS 2021 showed it in clathrin coated pits).*

We have now extended the discussion from **p 15, line 426** to cover this point: “*The implication is that, even though PI(4,5)P₂ itself can be an anchoring component of the membrane proteins, any unbound PI(4,5)P₂ can rapidly and freely diffuse away from the complex. To quote an example, a $\sim 100 \text{ nm}$ clathrin coated pit would lose an unbound PI(4,5)P₂ molecule from the complex within $\sim 26 \text{ ms}$, assuming diffusion at $0.3 \mu\text{m}^2/\text{s}$ (from $r = \sqrt{4Dt/\pi}$). Therefore, even*

localized PI(4,5)P₂ synthesis at such structures would rapidly lose any local enrichment unless the newly synthesized lipids were rapidly bound by effectors.”

4. Description of analyses that authors prefer not to carry out

1.2 - *2. All of the data are acquired in cells that have been constantly maintained in complete media. What are the effects of starvation and stimulation on PIP₂ behaviors as read out by these sensors?*

Total cellular levels of PI(4,5)P₂ are relatively stable and do not change appreciably in most systems that have been studied after periods of serum starvation. Likewise, aside from pharmacologic levels of activation of PLCβ-coupled receptors, levels of PI(4,5)P₂ do not change in response to stimulation of the majority of the processes studied here in HeLa cells. We therefore agree with reviewer 2 that *“repeating these experiments in starvation condition (Major comment #2) is kind of outside the scope of the present work.”*

1.6 - *In the absence of direct visualization of preexisting PIP₂-effector complexes, some immunofluorescence (or similar approach) would be useful to strengthen the conclusions regarding the order of PIP₂ vs effector accumulation at the surface, especially since this is the most bold claim in the work.*

We are unclear how such an experiment would work. Immunofluorescence of PI(4,5)P₂ is subject to the same caveats at the biosensor experiments – that is, the specificity of the antibody binding is derived from the headgroup interaction, which is blocked when effectors are bound to these specific head groups.

2.5 - *What is the molar ratio of expression level of Spectrin/Septin vs others like CLC/Vinculin/ESyt1? Is it possible that if the second set of molecules are expressed in equivalent amounts, those can also hinder the PI(4,5)P₂ diffusion?*

The HeLa proteome gives us a direct answer to this question (pmid: 26496610):

Protein	Copy number	Concentration (nM)
SPTBN1 (β-spectrin)	539,790	448
SEPT2 (Septin 2)	955,593	793
CLTA (clathrin light chain)	2,201,123	1,828
VCL (vinculin)	478,711	397
ESYT1	912,760	758

It is striking from these data that simple protein copy number/cellular concentration is not the reason for impeded diffusion of PI(4,5)P₂ by septins and spectrins in their native complexes. These are present at equivalent or lower concentrations than E-

Syt1 and clathrin light chain. However, given that most of these complexes were studied by tagging of the endogenous proteins in their native complexes, which present the proteins at appropriate local concentration, expression levels are somewhat moot. We therefore believe including this discussion in the manuscript would confuse.

May 12, 2022

Re: JCB manuscript #202204099T

Dr. Gerry R Hammond
University of Pittsburgh School of Medicine
Department of Cell Biology
BST-South, Room #327 3500 Terrace St
Pittsburgh, PA 15261

Dear Dr. Hammond,

Thank you for submitting your manuscript entitled "Free diffusion of PI(4,5)P2 in the plasma membrane in the presence of high density effector protein complexes". We apologize for the delay in communicating a decision to you. We have assessed your manuscript, the reviews, and your proposed revision plan. Your revision plan seems reasonable and we think the study would be of interest to a broad audience beyond phosphoinositide biologists. Therefore we would like to invite you to submit a revision if you can address the reviewer concerns.

GENERAL GUIDELINES:

Text limits: Character count for an Transfer is < 40,000, not including spaces. Count includes title page, abstract, introduction, results, discussion, and acknowledgments. Count does not include materials and methods, figure legends, references, tables, or supplemental legends.

Figures: Transfers may have up to 10 main text figures. Figures must be prepared according to the policies outlined in our Instructions to Authors, under Data Presentation, <https://jcb.rupress.org/site/misc/ifora.xhtml>. All figures in accepted manuscripts will be screened prior to publication.

*****IMPORTANT:** It is JCB policy that if requested, original data images must be made available. Failure to provide original images upon request will result in unavoidable delays in publication. Please ensure that you have access to all original microscopy and blot data images before submitting your revision. ***

Supplemental information: There are strict limits on the allowable amount of supplemental data. Transfers may have up to 5 supplemental figures. Up to 10 supplemental videos or flash animations are allowed. A summary of all supplemental material should appear at the end of the Materials and methods section.

Please note that JCB now requires authors to submit Source Data used to generate figures containing gels and Western blots with all revised manuscripts. This Source Data consists of fully uncropped and unprocessed images for each gel/blot displayed in the main and supplemental figures. Since your paper includes cropped gel and/or blot images, please be sure to provide one Source Data file for each figure that contains gels and/or blots along with your revised manuscript files. File names for Source Data figures should be alphanumeric without any spaces or special characters (i.e., SourceDataF#, where F# refers to the associated main figure number or SourceDataFS# for those associated with Supplementary figures). The lanes of the gels/blots should be labeled as they are in the associated figure, the place where cropping was applied should be marked (with a box), and molecular weight/size standards should be labeled wherever possible.

The typical timeframe for revisions is three to four months. While most universities and institutes have reopened labs and allowed researchers to begin working at nearly pre-pandemic levels, we at JCB realize that the lingering effects of the COVID-19 pandemic may still be impacting some aspects of your work, including the acquisition of equipment and reagents. Therefore, if you anticipate any difficulties in meeting this aforementioned revision time limit, please contact us and we can work with you to find an appropriate time frame for resubmission. Please note that papers are generally considered through only one revision cycle, so any revised manuscript will likely be either accepted or rejected.

Thank you for this interesting contribution to Journal of Cell Biology. You can contact us at the journal office with any questions, cellbio@rockefeller.edu or call (212) 327-8588.

Sincerely,

William Prinz
Monitoring Editor
Journal of Cell Biology

Lucia Morgado-Palacin, PhD
Scientific Editor
Journal of Cell Biology

Pacheco et al - Rebuttal

Manuscript number: RC-2022-01237

Corresponding author(s): Hammond, Gerald

1. General Statements

We are grateful to the reviewers for helpful suggestions and constructive feedback. We have now comprehensively amended the manuscript to address all of these comments in depth.

In our rebuttal, points are numbered X.Y, where X is reviewer and Y is point number (with minor points numbered sequentially after their major points). The rebuttal is split into (1) this section of general comments, (2) revisions incorporated in this version, (3) prior changes incorporated in response to the revision plan (as previously seen by the editor), and (4) justification for analyses we proposed not to perform

2. Revisions incorporated in this round

1.1 - The biggest concern in interpreting the data is results from the competitive binding of the lipid biosensors. Particularly in the context of measurements with co-expressed constructs, normal kinetics or measurements of cellular processes should be included. The simplest example is the kinetic of CME. Given the known dependence on PIP₂, any perturbation in this kinetic confounds the interpretation of the data substantially.

As we describe in the opening paragraph of the results, we believe that the biosensors, especially at the low levels they are expressed at for single molecule imaging, are unlikely to perturb physiologic function because such a small fraction of lipid is sequestered. Nonetheless, we agree with the reviewer that the manuscript is substantially strengthened by an empirical demonstration of this fact. We now include the following experiments, where the experimental parameters were tested in mock vs Tubby_c-mCherry expressing cells:

Domain	Experiment
E-Syt1 (ER:PM MCS)	E-Syt-1 puncta size and density
Clathrin light chain (clathrin coated structures)	Clathrin coated structure initiation rate and lifetime distribution.
Ezrin-EGFP (F-actin membrane binding)	Ezrin-EGFP morphology
sfGFP-Vinculin (focal adhesions)	sfGFP-vinculin-marked focal adhesion size and density
ARPC4-EGFP (Arp2/3)	ARPC4-EGFP puncta size and density
EGFP- β -spectrin (spectrin cytoskeleton)	EGFP- β -spectrin morphology
sfGFP-Septin2 (septin cytoskeleton)	sfGFP-Septin2 morphology

This is presented as a whole new **Figure S1**:

Pacheco et al - Rebuttal

Figure S1: Tubby_c-mCherry expression does not alter PM domain architecture. (A) examples of sfGFP-E-Syt1 edited cells, and the mean area and density of contact sites for 80 (control) or 88 (Tubby_c-mCherry) cells were measured. P values and statistics are given for the results of a Mann-Whitney U-test or Unpaired t-test as indicated. Bars show mean \pm 95% C.I. (B) Examples of sfGFP-CLTA edited cells, with the mean clathrin-coated structure initiation rate and lifetime cohorts of 10 (control) or 11 Tubby_c-mCherry expressing cells. Results of an unpaired t-test are indicated. Bars are mean \pm 95% C.I. (C) Representative examples of Ezrin-EGFP-expressing cells with and without Tubby_c-mCherry. (D) Examples of sfGFP-vinculin edited cells; mean vinculin-labelled adhesion area and density was measured in 55 (control) or 46 (Tubby_c-mCherry expressing) cells. P values and statistics are given for the results of a Mann-Whitney U-test or Unpaired t-test as indicated. Bars show means \pm 95% C.I. (E) Images of ARPC4-EGFP expressing cells with and without Tubby_c-mCherry. The density of ARPC4-labelled puncta were quantified in 34 cells from each group; P values and U value are given for the results of a Mann-Whitney U-test. (F) Representative images of EGFP- β -spectrin expressing or (G) Septin2-sfGFP edited cells, with or without Tubby_c-mCherry expression. Throughout, data were subject to a D'Agostino & Pearson normality tests to determine whether t- or U-tests were performed.

Pacheco et al - Rebuttal

We also refer to these experiments in the text as follows:

Results (p. 9, lines 250-255): *As we described in the previous section, expression of Tubby_c-mCherry is unlikely to sequester a substantial fraction of the unbound PI(4,5)P₂ and compete with PI(4,5)P₂-dependent macromolecular complexes, especially when expressed at low levels to facilitate single molecule detection. To verify that this was true, we compared such complexes in the presence and absence of Tubby_c-mCherry, revealing no apparent changes as shown in **Figure S1**.*

Results (p. 10, lines 272-274): *Endogenous E-Syt1 exhibits a punctate morphology, often strung along tubule-like distributions (**figure 5B**); the size and density of such structures were not altered by Tubby_c expression (**figure S1A**).*

Results (p. 11, lines 311-312): *In these cells, CCS appear as diffraction-limited spots; the density and dynamics of these spots are not altered by Tubby_c expression (**figure S1B**).*

Results (p. 12, lines 341-342), discussing Ezrin, vinculin, and Arp2/3: *There was no detectable change in the morphology of any of these structures with Tubby_c expression (**figure S1C-E**).*

Results (p. 13, lines 356-359): *In TIRFM, a largely amorphous but patchy distribution is observed (**figure 8C**), similar to previous observations in fibroblasts (Ghisleni et al., 2020). This distribution is not altered by Tubby_c expression (**figure S1F**).*

Results (p 13, lines 375-376): *Gross morphology of septin2-sfGFP was unchanged by Tubby_c expression (**figure S1G**).*

We are thus able to demonstrate that these membrane structures are unperturbed by biosensor expression.

1.4 - *Masking of spectrin data (Fig. 8), are not very compelling. A much more complete presentation of the data are required, especially to justify the strong conclusions.*

We agree with the reviewer that these data do not enable the reader to fully scrutinize the distribution of spectrin, and how diffusion changes at high sites of high spectrin density. In the revised manuscript, we have included the complete area of thresholded membrane (with an inset to show a representative overlaid trajectory), and amended the figure legend accordingly (bolded) for **Figure 8**:

2.3 - Why the authors preferred Tubby_c over PH-PLCδ for studies in complex structures is unclear. The argument of "Tubby_c behaves most similarly to the other lipid sensors" is weak as a PI(4,5)P₂ sensor does not necessarily need to behave similarly to a PI4P/PS sensor to be a better sensor. It would be interesting to see PH-PLCδ data in some of these structures.

Firstly, we have amended the first paragraph of this section of (A) the manuscript (p. 9, line 235) to clarify the selection of Tubby_c over PH-PLCδ1: "For these experiments, we elected not to employ the PH-PLCδ1, since we found in the previous section that it diffused significantly slower than other lipid biosensors, which can be ascribed to lipid-extrinsic protein-protein interactions (Hammond et al., 2009). We instead employed the Tubby_c PI(4,5)P₂ biosensor, since Tubby_c behaves most similarly to the other lipid sensors (figures 3, 4 and table 3). This is a good indication of unimpeded diffusion of the lipid:biosensor complex, since a lipid-selective biosensor interacts with the headgroup of the lipid, preventing lipid-selective interactions with other proteins that could impair diffusion for the duration of the biosensor complex. Diffusion should therefore be mainly limited by the viscous drag of the acyl chains in the membrane, which is not expected to differ significantly among different lipid classes (or the dually acylated Lyn₁₁ peptide). For Tubby_c, but not PH-PLCδ1, this seems to be the case."

Pacheco et al - Rebuttal

We did collect data with PH-PLC δ 1, with the same broad findings but much lower diffusion coefficients. However, we are of the opinion that including these data detracts from the main conclusions of the paper by introducing the confound of this lipid-extrinsic effect on diffusion.

2.4 - *Since the main claim of the paper is the unhindered PIP2 diffusion in the complex structures, it would be a great addition if, as a proof concept, at least one of the structures can be imaged in super resolution mode while doing the single molecule tracking in other channel, as that would show the finer details of the structures and demonstrate that the results were valid.*

This is a great experiment, though extremely technically challenging. The only super-resolution technique compatible with the high frame rates required for concomitant single particle tracking is photoactivated localization microscopy (PALM) – indeed, sptPALM is what we use for the tracking. However, obtaining adequate localizations to sample a structure at super resolution typically requires the collection of many millions of individual molecular localizations over a much longer time-frame than typically accomplished in our ~minute long spt tracking experiments.

We attempted these live-cell PALM-based approaches with a PAmCherry-tagged β -spectrin. Unfortunately, we are unable to generate a faithful super-resolution reconstruction of the underlying structure. A background-subtracted sum of the raw single molecule data generates a reconstruction of the underlying structure at the diffraction limit of resolution; the reconstruction of super-resolved localizations should look like a sharper version of this, and this image can be re-convolved to approximate the diffraction limited data. This can be demonstrated with some of our STORM data for microtubules, with reference to the original TIRF image:

Pacheco et al - Rebuttal

Unfortunately, with PAmCherry- β -spectrin, the reconstruction and convolved images introduce artifacts that do not faithfully reconstruct the diffraction limited sum of intensity:

Note, with PAmCherry, it is not possible to acquire a TIRF image since the fluor is not activated.

Therefore, we are unable to perform the requested experiment at this time; while the technical challenges may not be insurmountable, they are beyond the scope of reviewer revisions in terms of the time and resources required.

3. Revisions that were already incorporated in the transferred manuscript

1.3 - *For diffusional analyses, only trajectories longer than 0.88 s (16 frames) were used. This is never justified. All data could be fit using a step size distribution and may be informative, especially given the observations of two mobility populations.*

We thank the reviewer for spotting this omission. We have now amended the methods on **p 19, line 525** with the requested information: “*For all analyses, only those trajectories longer or equal to 0.88 s (16 frames) were used. This time frame was selected as a minimum to allow sufficient time lags for mean square displacement analysis of individual trajectories, including a robust analysis of alpha, as described next. For consistency’s sake, the same filtered data were used for that analysis and for the analysis of independent displacements.*”

Pacheco et al - Rebuttal

The reviewer is correct, we could have used all displacements for the step size distribution (which we performed by cumulative step size distribution), but we did not for consistency with the individual trajectory data. However, an analysis for Tubby_c comparing the data presented in figure 3 shows that the same result is obtained when the filtered data (≥ 0.88 s trajectory length) are used as when all data (i.e., ≥ 2 localizations in subsequent frames, separated by 55 ms) are used:

1.5 - In Figure 2C, there is evidence of bimodality in the diameter distribution. This should be discussed.

The reviewer has a keen eye! We have now added an explanation of this observation in the results on **pp 5-6, lines 129-133**: “We did note a minor peak of fluorescence spot size centered around the 65-75 nm bin (**figure 2C**), representing less than 2.5% of all localizations. Given our pixel size of 65 nm in image space, we believe this is caused by “hot pixels” in our sCMOS that locally enhance signal and lead to an artifactually narrower distribution in a few single molecule localizations.”

2.1 - The authors did not explain why they are considering that only “free” lipids are involved in their diffusion studies and why the population of “bound” lipids must be distinct from the “free” counterpart? Intuitively, when new PI(4,5)P2 molecules are synthesized on the membrane they can either bind to effectors or biosensors and also possibly dynamically interchange with both. In fact, as authors are possibly aware, in some cases, PH domains were shown to be hinder agonist induced hydrolysis of PI(4,5)P2 (Varnai, Balla JCB 1998) which suggests the pools of PI(4,5)P2 may be similar.

Pacheco et al - Rebuttal

We completely agree with the reviewer here, who has clearly articulated a key point of our interpretation that we failed to communicate in the opening paragraph of the results section and figure 1. Our mistake was to use the word “free” for a lipid, which has two meanings in our manuscript: (i) unhindered, rapid and Brownian diffusion of lipids and (ii) unbound lipids. We have now clarified by using the term “unbound” where we meant it, instead of “free”. Hopefully, the revised figure 1 and opening paragraph of the results on **pp. 4-5, lines 92-110** clarifies sufficiently to address the reviewer’s concern:

Figure 1: Lipid biosensors and pools of plasma membrane lipid. Functional membrane lipids such as PI(4,5)P₂ are expected to exist in a dynamic equilibrium between “unbound lipid” where the headgroup does not engage proteins, and “effector bound” pool where headgroup binds effector proteins. Biosensors like Tubby_c-PAmCherry (TBY) reversibly interact with and thus sample the “unbound” pool of lipid.

“PI(4,5)P₂ in the PM exists in a rapid dynamic equilibrium with its effector proteins (figure 1). Estimates using fluorescent acyl chain derivatives indicate that two out of three PI(4,5)P₂ molecules are in complex with such proteins at any given moment (Golebiewska et al., 2008). In this manuscript, we consider the remaining one third of lipid molecules that are estimated to be unbound. When new complexes are assembled, or new effector proteins are recruited to these complexes, it is this unbound lipid that recruits them; this is the lipid pool that must be locally concentrated to modulate an effector complex. We therefore measured the diffusion of these unbound lipid molecules. To this end, we used genetically encoded lipid biosensors, which interact with the headgroup. The biosensors themselves are in rapid dynamic equilibrium with the lipids and preclude interaction with endogenous effector

proteins (figure 1). Unlike effector proteins, the biosensors are estimated to sequester a much smaller fraction of PI(4,5)P₂, likely less than 10% (Wills et al., 2018). Although expression of high concentrations of biosensor can sequester a higher fraction of the unbound pool, reducing the pool available for effector interaction and inhibiting phospholipase C, for example (Várnai and Balla, 1998), this is unlikely to occur in the experiments described below, since low expression levels were utilized to favor resolution of single molecules. Crucially, previous studies have shown that the diffusion coefficient of biosensor-bound PI(4,5)P₂ is unchanged from the unbound lipid (Mashanov and Molloy, 2007; Yaradanakul and Hilgemann, 2007; Golebiewska et al., 2008; Hammond et al., 2009).”

2.2 - *The authors used different biosensor diffusion as a proxy of lipid diffusion. While that is a major established way of visualizing lipid diffusion, the authors need to explain how they accounted for different physiological processes that would cause the biosensors to be lost from the membrane? For example, as it was shown in different systems, membrane bound biosensors fall off to cytosol when a lipid is degraded. Additionally, biosensors often turnover very fast. For e.g. at single molecule level, PIP3 biosensor was shown to bind to PIP3 for only ~120 ms (Matsuoka et al. J Cell Sci 2006). Did the authors observe such fast on and off mechanism (shuttling) with any of biosensors in their system and if they did, how did they update their diffusion analysis?*

Pacheco et al - Rebuttal

We have now accounted for the effects of shuttling of the biosensors and enzymatic degradation of the lipid in a new paragraph in the results on **pp. 6-7 lines 134-148**: *“The single biosensor molecules represented complexes with lipids on the membranes imaged in TIRFM. They are readily discerned from unbound, cytoplasmic biosensors diffusing just above but in the plane of the membrane by the relative diffusion rates: lipids diffuse up to $\sim 1 \mu\text{m}^2/\text{s}$ in cells, whereas cytosolic biosensors diffuse around $20 \mu\text{m}^2/\text{s}$ (Hammond et al., 2009). Displacement of such molecules in the camera exposure time, $t = 55 \text{ ms}$, can be estimated from $\sqrt{4Dt/\pi}$ (Teruel and Meyer, 2000) at approximately 260 nm and 1.2 μm respectively. The large degree of displacement in the latter case “blurs” and spreads the intensity of the single molecule image to the extent that it is no longer resolvable against camera noise. It follows that biosensors can only be tracked for the lifetime that they stay in complex with the lipid. Estimates of the lifetime of Tubby_c molecules extrapolated back to zero laser-induced photobleaching place this binding lifetime at 339 ms (**figure 2E**). Enzymatic turnover of the lipids, or engagement with an endogenous effector protein requires the dissociation of the biosensor first, so these processes do not impact our diffusion measurements. Indeed, we have previously demonstrated that rates of biosensor-bound lipid catabolism are limited by the dissociate rate of the biosensor (Hammond et al., 2009).”*

2.6 - *PI(4,5)P2 is mentioned as “master regulator” in multiple places in the paper (e.g. line 22, 50), whereas the current understanding in the signaling and cytoskeletal research field is that no particular protein or lipid can be considered as the “master regulator” due to the existence of multiple redundancies, crosstalks, and feedback loops.*

We removed the term “master” from line 50 of the introduction, since it was redundant. We have left the term “a master regulator”, since this is different from “master regulator” or “the master regulator” as implied by the reviewer. In our context, we mean master regulator to mean shared, not necessarily the hierarchically most important. Essentially, like a master key that is able unlock most PM processes, though it is not necessarily the only point of regulation.

2.7 - *Quite a few images (and their insets) do not have scale bars.*

We have amended the figures so that they all include an indication of the inset size in the figure and legend; for many images, the inset box also serves as the scale information

2.8 - *The numbering in Figure 4 is not correct - the figure referred as figure 4C in line 193 and 194 actually remained unnumbered in figure.*

We thank the reviewer for spotting this error on our part. This is now fixed on **p. 8 lines 217-226**.

Pacheco et al - Rebuttal

2.9 - *The authors should delineate in the Discussion how different the lipid enrichment would be if PI(4,5)P₂ are locally synthesized in those special structures, as has been claimed (for e.g. Jung S-R et al. PNAS 2021 showed it in clathrin coated pits).*

We have now extended the discussion from **pp 15-16, lines 436-442** to cover this point: “The implication is that, even though PI(4,5)P₂ itself can be an anchoring component of the membrane proteins, any unbound PI(4,5)P₂ can rapidly and freely diffuse away from the complex. To quote an example, a ~100 nm clathrin coated pit would lose an unbound PI(4,5)P₂ molecule from the complex within ~26 ms, assuming diffusion at 0.3 μm²/s (from $r = \sqrt{4Dt/\pi}$). Therefore, even localized PI(4,5)P₂ synthesis at such structures would rapidly lose any local enrichment unless the newly synthesized lipids were rapidly bound by effectors.”

4. Description of analyses that authors prefer not to carry out

1.2 - *2. All of the data are acquired in cells that have been constantly maintained in complete media. What are the effects of starvation and stimulation on PIP2 behaviors as read out by these sensors?*

Total cellular levels of PI(4,5)P₂ are relatively stable and do not change appreciably in most systems that have been studied after periods of serum starvation. Likewise, aside from pharmacologic levels of activation of PLCβ-coupled receptors, levels of PI(4,5)P₂ do not change in response to stimulation of the majority of the processes studied here in HeLa cells. We therefore agree with reviewer 2 that “repeating these experiments in starvation condition (Major comment #2) is kind of outside the scope of the present work.”

1.6 - *In the absence of direct visualization of preexisting PIP2-effector complexes, some immunofluorescence (or similar approach) would be useful to strengthen the conclusions regarding the order of PIP2 vs effector accumulation at the surface, especially since this is the most bold claim in the work.*

We are unclear how such an experiment would work. Immunofluorescence of PI(4,5)P₂ is subject to the same caveats at the biosensor experiments – that is, the specificity of the antibody binding is derived from the headgroup interaction, which is blocked when effectors are bound to these specific head groups.

2.5 - *What is the molar ratio of expression level of Spectrin/Septin vs others like CLC/Vinculin/ESyt1? Is it possible that if the second set of molecules are expressed in equivalent amounts, those can also hinder the PI(4,5)P₂ diffusion?*

Pacheco et al - Rebuttal

The HeLa proteome gives us a direct answer to this question (pmid: 26496610):

Protein	Copy number	Concentration (nM)
SPTBN1 (β -spectrin)	539,790	448
SEPT2 (Septin 2)	955,593	793
CLTA (clathrin light chain)	2,201,123	1,828
VCL (vinculin)	478,711	397
ESYT1	912,760	758

It is striking from these data that simple protein copy number/cellular concentration is not the reason for impeded diffusion of PI(4,5)P₂ by septins and spectrins in their native complexes. These are present at equivalent or lower concentrations than E-Syt1 and clathrin light chain. However, given that most of these complexes were studied by tagging of the endogenous proteins in their native complexes, which present the proteins at appropriate local concentration, expression levels are somewhat moot. We therefore believe including this discussion in the manuscript would confuse.

September 19, 2022

RE: JCB Manuscript #202204099R

Dr. Gerry R Hammond
University of Pittsburgh School of Medicine
Department of Cell Biology
BST-South, Room #327 3500 Terrace St
Pittsburgh, PA 15261

Dear Dr. Hammond:

Thank you for submitting your revised manuscript entitled "Free diffusion of PI(4,5)P2 in the plasma membrane in the presence of high density effector protein complexes". The reviewers have now assessed your revised manuscript and, as you can see, they are overall satisfied with revisions. Reviewer #1, who acknowledges that the super resolution experiment they initially proposed was challenging, would like you to instead increase the size of the macromolecular structures by pharmacological approaches and perform diffusion analysis. However, in our view, this would not be necessary for final acceptance. Thus, we would be happy to publish your paper in JCB pending final revisions necessary to meet our formatting guidelines (see details below).

To avoid unnecessary delays in the acceptance and publication of your paper, please read the following information carefully. Please go through all the formatting points paying special attention to those marked with asterisks.

A. MANUSCRIPT ORGANIZATION AND FORMATTING:

Full guidelines are available on our Instructions for Authors page, <https://jcb.rupress.org/submission-guidelines#revised>.
Submission of a paper that does not conform to JCB guidelines will delay the acceptance of your manuscript.

1) Text limits: Character count for Articles and Tools is < 40,000, not including spaces. Count includes title page, abstract, introduction, results, discussion, and acknowledgments. Count does not include materials and methods, figure legends, references, tables, or supplemental legends.

2) Figures limits: Articles and Tools may have up to 10 main text figures.

Please note that main text figures should be provided as individual, editable files.

3) Figure formatting:

Molecular weight or nucleic acid size markers must be included on all gel electrophoresis.

*** Scale bars must be present on all microscopy images, including inset magnifications. Please include scale bars in main Figs. 2B, 5B, 5E, 6B, 7A-D, 8C and 9B.

Also, please avoid pairing red and green for images and graphs to ensure legibility for color-blind readers. If red and green are paired for images, please ensure that the particular red and green hues used in micrographs are distinctive with any of the colorblind types. If not, please modify colors accordingly or provide separate images of the individual channels.

4) Statistical analysis:

Error bars on graphic representations of numerical data must be clearly described in the figure legend.

The number of independent data points (n) represented in a graph must be indicated in the legend. Please, indicate whether N refers to technical or biological replicates (i.e. number of analyzed cells, samples or animals, number of independent experiments).

Statistical methods should be explained in full in the materials and methods in a separate section.

For figures presenting pooled data the statistical measure should be defined in the figure legends.

Please also be sure to indicate the statistical tests used in each of your experiments (both in the figure legend itself and in a separate methods section) as well as the parameters of the test (for example, if you ran a t-test, please indicate if it was one- or two-sided, etc.).

*** As you used parametric tests in your study (i.e. t-tests), you should have first determined whether the data was normally distributed before selecting that test. In the stats section of the methods, please indicate how you tested for normality. If you did not test for normality, you must state something to the effect that "Data distribution was assumed to be normal but this was not formally tested."

5) Abstract and title:

The abstract should be no longer than 160 words and should communicate the significance of the paper for a general audience.

*** The title should be less than 100 characters including spaces. Make the title concise but accessible to a general readership. Your title needs to be shortened, thus we would like to suggest something along the following lines: "PI(4,5)P2 diffuses freely in the plasma membrane within high density effector protein complexes"

6) Materials and methods:

*** Should be comprehensive and not simply reference a previous publication for details on how an experiment was performed. The text should not refer to methods "...as previously described."

Also, the materials and methods should be included with the main manuscript text and not in the supplementary materials.

7) Please be sure to provide the sequences for all of your primers/oligos and RNAi constructs in the materials and methods.

You must also indicate in the methods the source, species, and catalog numbers (where appropriate) for all of your antibodies. Please include species for all of your antibodies.

8) Microscope image acquisition:

The following information must be provided about the acquisition and processing of images:

a. Make and model of microscope

b. Type, magnification, and numerical aperture of the objective lenses

c. Temperature

*** d. imaging medium

e. Fluorochromes

f. Camera make and model

g. Acquisition software

h. Any software used for image processing subsequent to data acquisition. Please include details and types of operations involved (e.g., type of deconvolution, 3D reconstitutions, surface or volume rendering, gamma adjustments, etc.).

10) Supplemental materials:

There are strict limits on the allowable amount of supplemental data. Articles/Tools may have up to 5 supplemental figures. There is no limit for supplemental tables.

Please note that supplemental figures and tables should be provided as individual, editable files.

*** A summary of all supplemental material should appear at the end of the Materials and Methods section (please see any recent JCB paper for an example of this summary).

11) eTOC summary:

A ~40-50 word summary that describes the context and significance of the findings for a general readership should be included on the title page.

The statement should be written in the present tense and refer to the work in the third person. It should begin with "First author name(s) et al..." to match our preferred style.

12) Conflict of interest statement:

*** JCB requires inclusion of a statement in the acknowledgements regarding competing financial interests. If no competing financial interests exist, please include the following statement: "The authors declare no competing financial interests."

13) A separate author contribution section is required following the Acknowledgments in all research manuscripts.

*** All authors should be mentioned and designated by their first and middle initials and full surnames and the CRediT nomenclature is encouraged (<https://casrai.org/credit/>).

14) ORCID IDs: ORCID IDs are unique identifiers allowing researchers to create a record of their various scholarly contributions in a single place. At resubmission of your final files, please consider providing an ORCID ID for as many contributing authors as possible.

15) Materials and data sharing:

All animal and human studies must be conducted in compliance with relevant local guidelines, such as the US Department of Health and Human Services Guide for the Care and Use of Laboratory Animals or MRC guidelines, and must be approved by the authors' Institutional Review Board(s). A statement to this effect with the name of the approving IRB(s) must be included in the Materials and Methods section.

*** As a condition of publication, authors must make protocols and unique materials (including, but not limited to, cloned DNAs; antibodies; bacterial, animal, or plant cells; and viruses) described in our published articles freely available upon request by researchers, who may use them in their own laboratory only. All materials must be made available on request and without undue delay. Please, indicate whether the cell lines and reagents generated in this study have been deposited in public repositories. If not, please state that they would be made available to the scientific community upon request in the 'Data availability' section.

All datasets included in the manuscript must be available from the date of online publication, and the source code for all custom computational methods, apart from commercial software programs, must be made available either in a publicly available database or as supplemental materials hosted on the journal website. Numerous resources exist for data storage and sharing (see Data Deposition: <https://rupress.org/jcb/pages/data-deposition>), and you should choose the most appropriate venue based on your data type and/or community standard. If no appropriate specific database exists, please deposit your data to an appropriate publicly available database.

16) Please note that JCB now requires authors to submit Source Data used to generate figures containing gels and Western blots with all revised manuscripts. This Source Data consists of fully uncropped and unprocessed images for each gel/blot displayed in the main and supplemental figures. The Source Data files will be directly linked to specific figures in the published article.

Since your paper includes cropped gel and/or blot images, please be sure to provide one Source Data file for each figure that contains gels and/or blots along with your revised manuscript files. File names for Source Data figures should be alphanumeric without any spaces or special characters (i.e., SourceDataF#, where F# refers to the associated main figure number or SourceDataFS# for those associated with Supplementary figures). The lanes of the gels/blots should be labeled as they are in the associated figure, the place where cropping was applied should be marked (with a box), and molecular weight/size standards should be labeled wherever possible.

B. FINAL FILES:

-- High-resolution figure and MP4 video files: See our detailed guidelines for preparing your production-ready images,

<https://jcb.rupress.org/fig-vid-guidelines>.

Thank you for your attention to these final processing requirements. Please revise and format the manuscript and upload materials within 7 days. Please let us know if any complication preventing you from meeting this deadline arises and we can work with you to determine a suitable revision period.

Thank you for this interesting contribution, we look forward to publishing your paper in Journal of Cell Biology.

Sincerely,

William Prinz
Monitoring Editor
Journal of Cell Biology

Lucia Morgado-Palacin, PhD
Scientific Editor
Journal of Cell Biology

Reviewer #1 (Comments to the Authors (Required)):

The authors have carried out or attempted to carry out most of the suggestions we have made. We understand that the super resolution experiment we suggested was extremely difficult. In lieu of that, could the authors apply some pharmacological/CID treatments that would increase the sizes of of any of the structures and then carry out the diffusion analysis across these enlarged structures (sorry we did not think of this initially)?

Reviewer #2 (Comments to the Authors (Required)):

Pacheco, et al combine live cell imaging and photoactivated single particle tracking (sptPALM) to measure spatial and temporal features PIP2 diffusion in the plasma membrane of HeLa cells. PIP2 binding domains are expressed and visualized in TIRF. The work, including the revisions, is clearly presented and demonstrates substantial effort. The clarifications in interpretations and additional data have greatly improved the manuscript.

Minor comments

1. There are some typos and the term 'master regulator' has been incompletely expunged (Abstract).